# Addressing Attribute Bias with Adversarial Support-Matching

**Thomas Kehrenberg**                                           *t.kehrenberg@sussex.ac.uk*
*Predictive Analytics Lab (PAL), University of Sussex*

**Myles Bartlett**                                              *m.bartlett@sussex.ac.uk*
*Predictive Analytics Lab (PAL), University of Sussex*

**Viktoriia Sharmanska**                                        *sharmanska.v@sussex.ac.uk*
*Predictive Analytics Lab (PAL), University of Sussex*

**Novi Quadrianto**                                             *n.quadrianto@sussex.ac.uk*
*Predictive Analytics Lab (PAL), University of Sussex*
*BCAM Severo Ochoa Strategic Lab on Trustworthy Machine Learning*
*Monash University, Indonesia*

**Reviewed on OpenReview:** *https://openreview.net/forum?id=JYbnJ92TJf*

## Abstract

When trained on diverse labelled data, machine learning models have proven themselves to be a powerful tool in all facets of society. However, due to budget limitations, deliberate or non-deliberate censorship, and other problems during data collection, certain groups may be under-represented in the labelled training set. We investigate a scenario in which the absence of certain data is linked to the second level of a two-level hierarchy in the data. Inspired by the idea of protected attributes from algorithmic fairness, we consider generalised secondary "attributes" which subdivide the classes into smaller partitions. We refer to the partitions defined by the combination of an attribute and a class label, or leaf nodes in aforementioned hierarchy, as *groups*. To characterise the problem, we introduce the concept of classes with *incomplete attribute support*. The representational bias in the training set can give rise to spurious correlations between the classes and the attributes which cause standard classification models to generalise poorly to unseen groups. To overcome this bias, we make use of an additional, diverse but unlabelled dataset, called the *deployment set*, to learn a representation that is invariant to the attributes. This is done by adversarially matching the support of the training and deployment sets in representation space using a set discriminator operating on sets, or *bags*, of samples. In order to learn the desired invariance, it is paramount that the bags are balanced by class; this is easily achieved for the training set, but requires using semi-supervised clustering for the deployment set. We demonstrate the effectiveness of our method on several datasets and realisations of the problem. Code for the paper is publicly available at `https://github.com/wearepal/support-matching`.

## 1 Introduction

Machine learning has burgeoned in the last decade, showing the ability to solve a wide variety of tasks with unprecedented accuracy and efficiency. These tasks range from image classification (Krizhevsky et al., 2012) and object detection (Ren et al., 2015), to recommender systems (Ying et al., 2018) and the modelling of complex physical systems such as precipitation (Ravuri et al., 2021) and protein folding (Jumper et al., 2021). In the shadow of this success, however, one finds less cause for optimism in frequent failure in equitability and generalisation-capability, failure which can have serious repercussions in high-stakes applications such as self-

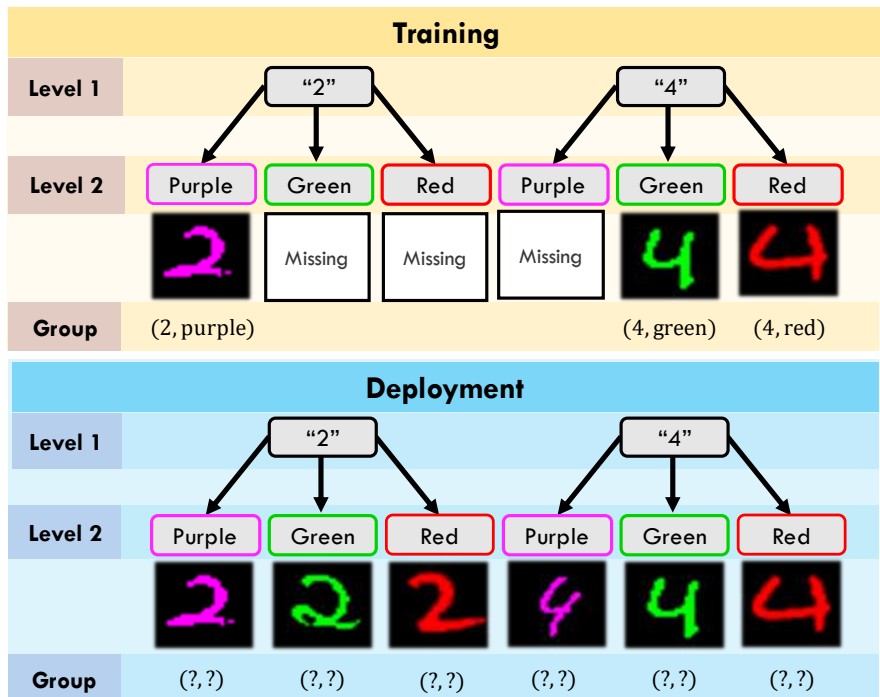

Figure 1: Illustration of our general problem setup. We assume the data follows a two-level hierarchy in which the first level corresponds to the class-level information (digit) and the second level being defined by the secondary attribute (colour). While all digits appear in the training set (Top), not all digit-colour combinations (groups) do; these gaps in class-conditional support give rise to a spurious correlation between digit and colour, where the former is completely determined by the latter in the training set (giving the mappings purple → 2 and green ∨ red → 4 as degenerate solutions to the classification problem), yet this same correlation does not hold for the deployment set (Bottom) which contains samples from the missing combinations. To disentangle the (spurious) attribute- and class-related information, we make use of an additional dataset that is representative of the data the model is expected to encounter at deployment time, in terms of the groups present.

driving cars (Sun et al., 2019), judicial decision-making (Mayson, 2018), and medical diagnoses (AlBadawy et al., 2018). ML's data-driven nature is a double-edged sword: while it opens up the ability to learn patterns that are infeasibly complex for a practitioner to encode by hand, the quality of the solutions learned by these models depends primarily on the quality of the data with which they were trained. If the practitioner does not properly account for this, models ingesting data ridden with biases will assimilate, and sometimes even amplify, those biases. The problem boils down to not having sufficiently diverse annotated data. However, collecting more labelled data is often infeasible due to temporal, monetary, legal, etc., constraints.

While data can be intrinsically biased (such as in the case of bail records), *representational bias* is more often to blame, where socioeconomic or regulatory factors resulting in certain demographics being under- (or even un-) represented. Clinical datasets are particularly problematic for ML due to the frequency of the different outcomes being naturally highly imbalanced, with the number of negative cases (`healthy`) typically greatly outweighing the number of positive cases (`diseased`); even if a demographic is well-represented overall, that may well not be the case when conditioned on the outcome. For example, pregnant women are often excluded from clinical trials due to safety concerns; when they do participate it is often at rates that are too low to be meaningful (Afrose et al., 2021).

Like many prior works (Sohoni et al., 2020; Kim et al., 2019; Creager et al., 2021; Sagawa et al., 2020), we consider settings where there is a two-level hierarchy, with the second level partitioning the data based on secondary *attributes* that are causally independent of the class (constituting the first level) which is being

predicted. This second level of the data is assumed to be predictable by the classifiers in the considered hypothesis class. In both Sohoni et al. (2020) and Creager et al. (2019) the entailed secondary attributes are unobserved and need to be inferred in a semi-supervised fashion. We consider a similar problem but one where the second level is partially observed. Specifically, we focus on problems where some outcomes are available for some attributes and not for others. This particular form of the problem has – so far as we are aware – been hitherto largely overlooked despite pertaining to a number of real-world problems, only having recently garnered attention in the concurrent work of Yang et al. (2023).

If the labelled training set is sufficiently balanced in terms of classes and attributes, a standard ERM (empirical risk minimisation) classifier can achieve good performance. However, we consider the added difficulty that, in the labelled training set, some outcomes (classes) are not observed for all attributes, meaning some of the classes do not overlap with all the attributes. In other words, in the training set, some of the classes have *incomplete support* with respect to the partition given by the attributes, while in the deployment setting we expect all possible combinations of attribute and class to appear. We illustrate our problem setup in Fig. 1, using Coloured MNIST digits as examples; here, the first level of the hierarchy captures digit class, the second level, colour. While the (unlabelled) deployment set contains all digit-colour combinations (or *groups*), half of these combinations are missing from the (labelled) training set. A classifier trained using only this labelled data would wrongly learn to classify 2's based on their being purple and 4's, based on their being green or red (instead of based on shape) and when deployed would perform no better than random due to the new groups being coloured contrary to their class (relative to the training set).

We address this problem by learning representations that are invariant to the attribute and that thus enable the model to ignore the attribute partition and to predict only the class labels. In order to train an encoder capable of producing these representations, the information contained in the labelled training set alone is not sufficient to break the *spurious correlations*. To learn the "correct" representations, we make use of an additional unlabelled dataset with support equivalent to that of the deployment set (which includes the possibility of it being the actual deployment set). We do not consider this a significant drawback as such data is almost always far cheaper to procure than *labelled* data (which may require expert knowledge).

This additional dataset serves as the inductive bias needed by the encoder to disentangle class- and attribute-related factors. The encoder is trained adversarially to produce representations whose source (`training` or `deployment`) is indeterminable to a set-classifier. To ensure attribute- (not class-) invariance is learned, the batches fed to the discriminator need to be approximately balanced, such that they reflect the support, and not the shape, of the distributions.

We empirically show that our proposed method can effectively disentangle attribute-related and semantic factors on a range of visual, classification datasets and is robust to noise in the bag-balancing, to the degree of outperforming the baseline methods even when no balancing of bags from the deployment set is performed. Furthermore, we prove that the entailed objective is in the limit guaranteed to yield representations that are invariant to attributes and that we can bound the error incurred due to imperfect clustering.

## 2 Problem setup

In this section, we illustrate and formalise the problem of classes with incomplete attribute-support. We start by defining requisite notation for conveying our setup and in Sec. 2.2 expand on this notation to construct a more general and compact description of said problem. Let $x \in \mathcal{X} \subset \mathbb{R}^d$, $y \in \mathcal{Y}$ and $a \in \mathcal{A}$ denote the observed input features, class labels and attribute labels, respectively, with $\mathcal{Y}$ and $\mathcal{A}$ being non-empty, finite sets, and with upper-case letters denoting observed variables' random-variable counterparts here ($X$, $Y$, and $A$) and throughout. We refer to the values, $g \in \mathcal{G}$ as *groups*, representing unique pairs of $a$ and $y$, such that $\mathcal{G} \subseteq \mathcal{A} \times \mathcal{Y}$. As in a standard supervised learning task, we have access to a labelled training set $\mathcal{D}^{tr} \triangleq \{(x_n, a_n, y_n)\}_{n=1}^{N^{tr}} \subset (\mathcal{X} \times \mathcal{A} \times \mathcal{Y})$, that is used to train a classifier $\Gamma : \mathcal{X} \to \mathcal{Y}$ that is then deployed on test set $\mathcal{D}^{te} \triangleq \{(x_n, a_n, y_n)\}_{n=1}^{N^{te}} \subset (\mathcal{X} \times \mathcal{A} \times \mathcal{Y})$. We use superscript, to denote association of a domain with a correspondingly superscripted dataset, e.g. $\mathcal{G}^{tr}$ and $\mathcal{G}^{te}$ respectively denote the groups present in the training and test sets. Lastly, for some functions, we abuse notation and allow the random and observed

variables to be interchanged as inputs. For example, given function $f : \mathcal{X} \to \mathbb{R}$, we may write both $f(x)$ and $f(X)$, the latter being a random variable itself.

## 2.1 Spurious correlations from missing groups

The spurious correlation (SC; Arjovsky et al., 2019), or shortcut-learning (Valle-Perez et al., 2018; Geirhos et al., 2020), problem is characterised by the presence of some secondary attribute $a$ (such as background (Beery et al., 2018), texture (Geirhos et al., 2018), or gender (Sagawa et al., 2019; Seyyed-Kalantari et al., 2020) that confounds the prediction task. In fair machine learning (FairML; Barocas et al., 2019), this secondary attribute corresponds to a "sensitive attribute" or "protected attribute" (sometimes denoted by '$s$'). The attribute is strongly correlated with the target label, $y$, in the training set, but spuriously so in the sense that the mapping $\mathcal{A} \to \mathcal{Y}$ is acausal and thus cannot be expected to hold at deployment time. This correlation is pernicious when $A$ is of lower complexity (which can be formalised in the Kolmogorov sense; Scimeca et al. (2021)) than the causal cues contained in $X$, and thereby becomes the preferred cue by virtue of simplicity bias (Valle-Perez et al., 2018). Such problems have garnered considerable attention in recent years (Liu et al., 2021; Pezeshki et al., 2021; Sohoni et al., 2020; Krueger et al., 2021) due to their pervasive, and potentially catastrophic (Codevilla et al., 2019; De Haan et al., 2019; Castro et al., 2020), nature. In this paper, we introduce, and propose a semi-supervised solution for, a hierarchical and class-asymmetric variant of the SC problem that is characterised by *missing groups*.

To illustrate the general SC problem and the version of it induced by missing groups, we define the conditional-probability matrix, $\mathbf{P}^{tr} \in [0,1]^{|\mathcal{A}| \times |\mathcal{Y}|}$, where each element $\mathbf{P}^{tr}_{ij}$ encodes the conditional probability $P^{tr}(Y = j | A = i)$ in the training set, $\mathcal{D}^{tr}$. When $\mathbf{P}^{tr}$ is both binary and doubly stochastic (that is, has all rows and columns summing to 1), we have that $y$ is completely determined by $a$ in $\mathcal{D}^{tr}$. This is an extreme form of the SC problem which is statistically intractable without access to additional sources of data (Kehrenberg et al., 2020) or multiple environments (Arjovsky et al., 2019). The problem of missing groups can be viewed as a relaxation of this SC problem wherein the elements of $\mathbf{P}^{tr}$ respect the constraint that all columns contain at least one non-zero value, i.e. we observe all class labels but not all possible pairs of class and attribute labels – we say that we have *missing groups*, $\mathcal{M} \triangleq \mathcal{G}^{te} \setminus \mathcal{G}^{tr}$. This setup still leads to spurious correlations but ones that are statistically tractable due to asymmetry. Practically speaking, considering only cases where groups are entirely missing is overly restrictive, and as such we instead view the problem setup as extending to cases where groups may not be altogether missing but have sample sizes too small to constitute meaningful supervision. To understand the non-triviality of this problem, and why aiming for invariance to $a$ in the training set alone – as is characteristic of many representation-learning methods in FairML (Edwards & Storkey, 2015; Madras et al., 2018; Quadrianto et al., 2019) and domain adaptation (DA; Ganin et al., 2016; Zhao et al., 2018) – will assuredly fail, consider a binary classification problem with a binary attribute, where $\mathcal{Y} = \{0, 1\}$ and $\mathcal{A} = \{0, 1\}$ and for which $\mathbf{P}^{tr}$ takes the form

$$
\mathbf{P}^{tr} = \begin{array}{c} \\ A = 0 \\ A = 1 \end{array} \overset{\begin{array}{cc} Y = 0 & Y = 1 \end{array}}{\left( \begin{array}{cc} 0.5 & 0.5 \\ 1.0 & 0.0 \end{array} \right)}. \tag{1}
$$

This represents a special case of the problem of missing groups that we refer to as the *Attribute Bias* (AB) problem, distinguished by the fact that we observe all attributes in at least one class, and corresponds to the *attribute generalisation* problem introduced in Yang et al. (2023). For example, we might have samples from $A = 0$ evenly distributed across both the negative and positive classes; for $A = 1$, however, we only observe samples from the negative class. This setup might appear somewhat benign at first blush, given that all classes are present in the training set, however, the fact that $a$ serves as a proxy for $y$ in the case of $A = 1$ frustrates our goal of attribute-invariant classification. The reason for this becomes obvious when decomposing a classifier into a mixture of experts (MoE), where $a$ indicates which expert to choose for the given sample. Such a model naturally arises in practice due to the tendency of deep neural networks to strongly favour shortcut solutions (Geirhos et al., 2020). We note that for this, and throughout the paper, we assume that $a$ is inferable, to some extent, from $x$, that is $\mathcal{I}(X; A) > 0$, with $\mathcal{I}(\cdot; \cdot)$ denoting the mutual information between two variables – this is almost always the case in practice but we make the dependence explicit here by denoting by $X_Y$ the causally-relevant component of $X$, that is independent of $A$, and by

including $a \in \{0, 1\}$ explicitly in the set of inputs. With this noted, we may then define the MoE classifier, $c_{MoE}$, that 'solves' the training set with labels distributed according to $\mathbf{P}^{tr}$ as

$$c_{MoE}(X_Y, A) = \begin{cases} c_{A=0}(X_Y) & \text{if } A = 0 \\ 0 & \text{if } A = 1 \end{cases}, \tag{2}$$

using $c_{A=0}(\cdot)$ to denote the expert that learns to classify only the subset of the data for which $A = 0$. Such a classifier is clearly undesirable, as should it ever encounter a sample with attribute 1 with a positive class label, the classifier will automatically declare it negative without needing to attend to $X_Y$ – it is invariant to $X_Y$ while being variant to $A$, which is the opposite of what we desire. Solutions to this problem often proceed by learning an encoder $f : \mathcal{X} \to \mathcal{Z}$ that maps an input $x$ into a representation, $z \in \mathcal{Z} \subset \mathbb{R}^l$, which has the desired property of $A$-invariance, $Z \perp A$, while also maximising $\mathcal{I}(Z; Y)$ so that the representation is useful for classification. A popular way of imparting this invariance is with adversarial methods (Ganin et al., 2016) where $f$ is trained to the equilibrium point, $f^*$, of the minimax equation

$$\min_{f \in \mathcal{F}} \max_{h \in \mathcal{H}} \mathbb{E}_{(x,a,y) \sim \mathcal{D}^{tr}} [\overbrace{h(f(x))_a}^{\text{invariance}} - \underbrace{\lambda \mathcal{I}(f(x); y)}_{\text{classification}}], \tag{3}$$

where $h \in \mathcal{H} \subseteq (\triangle(|\mathcal{A}|))^{\mathcal{Z}}$ is a parametric adversary with the codomain of the standard simplex over $\mathcal{A}$, and $\lambda \in \mathbb{R}^+$ is a positive scalar controlling the trade-off between the two constituent objectives. Under ideal conditions, when all possible pairs of $a$ and $y$ are observed, $f^*$ corresponds to the point at which $h$ is maximally entropic and occurs when $Z$ is invariant to $A$, and only $A$, while mutual information w.r.t. $Y$ is jointly maximised – from an optimisation standpoint, the gradients of first and second objectives are non-conflicting (i.e. have non-negative inner products; Yu et al., 2020) and there is no trade-off. However, in cases where we have missing groups, the waters are muddied: satisfying the first part of the objective connotes invariance not only to $A$, but also to $Y$, since $A$ can be predicted from $Y$ with above-random accuracy due to the skewed statistics of the dataset. This is patently problematic as $Y$ is the very thing we wish to predict and achieving invariance to $A$ does little good if our classifier can no longer utilise features predictive of $Y$.

Since we cannot achieve optimality for the competing invariance and classification terms simultaneously, we instead have a set of Pareto optimal solutions that collectively make up the Pareto front. This problem could be tackled with multi-objective optimisation (MOO; Deb, 2014). Specifically, Eq. 3, with $\lambda$ controlling the preference direction, characterises the most straightforward approach to MOO, called *linear scalarisation* (Boyd et al., 2004). MOO has recently been explored in the context of unsupervised domain adaptation (UDA), where gradients of alignment and classification terms conflict, (Liang et al., 2021), and in the context of FairML for controlling the trade-off between predictive performance and fairness (Navon et al., 2020). While our missing-groups problem admits a MOO-based approach, we are instead interested in leveraging unlabelled data to sidestep the implied trade-off altogether.

## 2.2 Formalising the problem

In order to provide a general formulation of the problem of missing groups exemplified above, we begin by defining additional notation for reasoning over label-conditioned subsets and their support. For a given dataset, $\mathcal{D}$, we denote by $\mathcal{D}_{A=a}$ its subset with attribute label $a \in \mathcal{A}$, by $\mathcal{D}_{Y=y}$ its subset with class label $y \in \mathcal{Y}$, and – combining the two – by $\mathcal{D}_{A=a,Y=y}$ its subset with attribute label $a$ and class label $y$. According to this scheme, $\mathcal{D}_{A=\text{purple},Y=2}$ should then be read as "the set of all samples in $\mathcal{D}$ with class label '2' and attribute label 'purple' ". We apply similar syntax to the partitions induced by the attribute, writing $\mathcal{A}^{tr}_{Y=y}$ to mean the observed attributes within class $y$ in the training set. For instance, $\mathcal{A}^{tr}_{Y=1} = \{0\}$ prescribes that for class 1, only attribute 0 is observed in the training set.

We assume a problem of a hierarchical nature of the kind illustrated in Fig. 1. While the full set of class labels is observed in both the training and test sets, we do not observe all pairs of $a$ and $y$ in the former, i.e. $\mathcal{G}^{tr} \subsetneq \mathcal{G}^{te}$ or $\mathcal{M} \neq \varnothing$. Equivalently, we say that for some class, $y^\dagger$, we have $\mathcal{A}^{tr}_{Y=y^\dagger} \subsetneq \mathcal{A}$, subject to

the constraint that $\mathcal{A}^{tr} = \mathcal{A}^{te}$ (i.e., all attributes are observed in some way). With this, we can succinctly notate the AB problem realised by Eq. 1, in which class $Y = 1$ has no overlap with attribute $A = 1$, as $\mathcal{A}^{tr}_{Y=1} = \{0\}$ (while $\mathcal{A}^{tr}_{Y=0} = \{0, 1\}$), corresponding to $\mathcal{M} = \{(1, 1)\}$, and distinguish AB problems generally by the inclusion of the additional constraint $\mathcal{A}^{tr} = \mathcal{A}^{te}$. To illustrate a more complex case, the AB problem depicted in Fig. 1, for which we observe exclusively purple '2's and green and red '4's, can be notated with the pair $\mathcal{A}^{tr}_{Y=2} = \{\text{purple}\}$, $\mathcal{A}^{tr}_{Y=4} = \{\text{green}, \text{red}\}$. Since we only deal with the AB setting in this paper, we will use the terms "AB problem" and "missing groups problem" synonymously hereafter.

### 2.3 A way forward

In this paper, we propose to alleviate the AB problem by mixing labelled data with *unlabelled* data that is usually much cheaper to obtain (Chapelle et al., 2006), referring to this set of *unlabelled* data as the *deployment set* [1], $\mathcal{D}^{dep}_{\star} = \{(x_n, a^{\star}_n, y^{\star}_n)\}^{N^{dep}}_{n=1} \subset (\mathcal{X} \times \mathcal{A} \times \mathcal{Y})$, using "$\star$" to denote that the labels are *unobserved*, and in practice we only have access to $\mathcal{D}^{dep} \triangleq \{(x_n)\}^{N^{dep}}_{n=1} \subseteq \mathcal{X}$ and must estimate the corresponding groups. We assume that this deployment set is group-complete w.r.t. the test set, $\mathcal{G}^{dep} = \mathcal{G}^{te}$. Leveraging this deployment set, we seek to learn a classifier, $\Gamma$, that can generalise well to the missing groups appearing in the test set without seeing any labelled representatives in the training set. In practice, we treat $\Gamma$ as a composition, $c \circ f$, of two subfunctions: an encoder $f : \mathcal{X} \to \mathcal{Z}$, which maps a given input $x$ to a representation $z \in \mathcal{Z} \subseteq \mathbb{R}^l$, and a classifier head $c : \mathcal{Z} \to \mathcal{Y}$, which completes the mapping to the space of class labels, $\mathcal{Y}$. Since the task of achieving independence between the predictions and attribute labels can be reduced to the task of learning the invariance $Z \perp A$; we next discuss how one can learn an encoder satisfying this condition in a theoretically-principled manner.

## 3 Adversarial Support-Matching

We cast the problem of learning an attribute-invariant representation as one of *support-matching* between a dataset that is *labelled* but has *incomplete* support over groups, $G$, and one, conversely, that has *complete* support over $G$, but is *unlabelled*. The idea is to produce a representation that is invariant to this difference in support, and thus invariant to the attribute. However, it is easy to learn the wrong invariance if one is not careful. To measure the discrepancy in support between the two distributions, we adopt an adversarial approach, but one where the adversary is operating on small sets – which we call *bags* – instead of individual samples. These bags need to be balanced with respect to $(a, y)$, such that we can interpret them as approximating $\mathcal{G}$ as opposed to the joint probability distribution, $P(A, Y)$. Details on how these bags are constructed can be found in Sec. 3.2 and Sec. 3.3.

### 3.1 Objective

We now present our overall support-matching objective. As alluded to before, the goal is to learn an encoder, $f$, which preserves all information relating to $Y$, but is invariant to $A$. Let $p_f(Z = z) \triangleq P^{tr}(f(X) = z)$ be the distribution resulting from sampling $x$ from $P^{tr}$ – the training set – and then transforming $x$ with $f$. Analogously for the deployment set: $q_f(Z = z) \triangleq P^{dep}(f(X) = z)$. Naïvely, we would want $p_f \stackrel{!}{=} q_f$, such that the output of the encoder is invariant to the source distribution, but we have to handle the fact that the training set does not contain all groups, which means $p_f$ necessarily has a different structure than $q_f$. The solution to this problem is to enforce $p_f \stackrel{!}{=} q_f$ for each group individually, by conditioning on $A$ and $Y$. For the conditioned distributions we write $p_f|_{A=a,Y=y}$ and $q_f|_{A=a,Y=y}$. However, now another problem appears: if the class $y$ has incomplete $\mathcal{A}$-support, then for some attribute $a \in \mathcal{A}$, $p_f|_{A=a,Y=y}$ will not be defined, because the group $(a, y)$ is missing from the training set. To deal with this, we introduce a *projection function* $\Pi$, which maps $(a, y)$ to a set of feasible attributes (i.e., attributes for which there is data in the

---

[1]In our experiments, we report accuracy and bias metrics on another independent test set instead of on the unlabelled data that is available at training time.

training set):

$$\Pi(a, y) = \begin{cases} \{a\} & \text{if } \mathcal{A}^{tr}_{Y=y} = \mathcal{A} \quad (\text{i.e., } y \text{ has full } \mathcal{A}\text{-support}) \\ \mathcal{A}^{tr}_{Y=y} & \text{otherwise.} \end{cases} \tag{4}$$

Intuitively, $\Pi(a, y)$ tells us how to sample data points such that the adversarial learning will learn the correct invariance – an invariance to $A$ but not to $Y$. For example, if $\mathcal{A}^{tr}_{Y=2} = \{\text{red}\}$, then $\Pi(\text{red}, 2) = \{\text{green}, \text{red}\}$. Instead of conditioning on $A = a$, we now condition on $A \in \Pi(a, y)$, meaning that the random variable $A$ may take on any of the values in the set $\Pi(a, y)$: $(A = a' \vee A = a'' \vee \dots)$ if $\Pi(a, y) = \{a', a'', \dots\}$. Our objective to minimize is then

$$\mathcal{L}_{\text{match}}(f) = \sum_{A \in \mathcal{A}} \sum_{y \in \mathcal{Y}} d(p_f|_{A \in \Pi(a,y), Y=y}, q_f|_{A=a, Y=y}) \tag{5}$$

where $d(\cdot, \cdot)$ is a distance measure for probability distributions. The optimal encoder $f^*$ is found by solving the following optimisation problem:

$$f^* = \underset{f \in \mathcal{F}}{\arg\min} \, \mathcal{L}_{\text{match}}(f) - \mathcal{I}(f(X); X) \tag{6}$$

where $\mathcal{I}(\cdot; \cdot)$ again denotes the mutual information. As written, Eq. 5 requires knowledge of $a$ and $y$ on the deployment set for conditioning. That is why, in practice, the distribution matching is not done separately for all combinations of $a \in \mathcal{A}$ and $y \in \mathcal{Y}$. Instead, we compare *bags* that contain samples from all combinations in the right proportions. For the deployment set, Eq. 5 implies that all $a$-$y$-combinations have to be present at the same rate in the bags, but for the training set, we need to implement $\Pi(a, y)$ with hierarchical balancing.

As the implications of the given objective might not be immediately clear, we provide the following proposition. The proof can be found in Appendix E.

**Proposition 1.** *If $f$ is such that*

$$p_f|_{A \in \Pi(a,y), Y=y} = q_f|_{A=a, Y=y} \quad \forall a \in \mathcal{A}, y \in \mathcal{Y} \tag{7}$$

*and $P^{tr}$ and $P^{dep}$ are data distributions that correspond to the real data distribution $P$, except that some $a$-$y$-combinations are less prevalent, or, in the case of $P^{tr}$, missing entirely, then, for every $y \in \mathcal{Y}$, there is either full coverage of $A$ for $y$ in the training set ($\mathcal{A}^{tr}_{Y=y} = \mathcal{A}$), or the following holds:*

$$P(A = a | f(X) = z, Y = y) = \frac{1}{n_a} \ . \tag{8}$$

*In other words: for $Y = y$, $f(x)$ is not predictive of $A$.*

## 3.2 Implementation

The implementation of the above objective combines elements from unsupervised representation-learning and adversarial learning. In addition to the invariant representation $z$, our model also outputs $\tilde{a}$, in a similar fashion to Kehrenberg et al. (2020) and Creager et al. (2019). This can be understood as a reconstruction of the attribute $a$ from the input $x$ and is necessary to prevent $z$ from being forced to encode $a$ by the reconstruction loss. This need could potentially be obviated using self-supervised approaches, but we abstain from this avenue in the interest of simplicity.

The model, $\Gamma$, is composed of three core modules: 1) two *encoder* functions, $f$ (which we refer to as the "debiaser") and $t$, which share weights and map $x$ to $z \in \mathcal{Z}$ and $\tilde{a} \in \tilde{\mathcal{A}}$, respectively; 2) a *decoder* function $r : \mathcal{Z} \times \tilde{\mathcal{A}} \to \mathcal{X}$ that learns to invert $f$ and $t$; and 3) a *discriminator* function $h : (\mathfrak{Z} \subseteq \mathcal{P}(\mathcal{Z})) \to (0, 1)$ that predicts which dataset a bag of samples, $\mathcal{B} \in \mathfrak{Z}$, embedded in $\mathcal{Z}$, was sampled from, where we have used $\mathcal{P}(\cdot)$ to denote the powerset of its argument and thereby a domain comprising sets of elements of $\mathcal{Z}$. The encoder $f$ is tasked with learning a representation $z$ such that it is indeterminate to the adversary $h$ whether a given bag originated from the deployment set ('positive') or the training set ('negative'). Formally, given

bags $\mathcal{B}^{tr}$, sampled according to $\Pi$ from the training set, and balanced bags from the deployment set, $\mathcal{B}^{dep}$, we first define, for convenience, the loss w.r.t. to the encoder networks, $f$ and $t$ as

$$\mathcal{L}_{\text{enc}}(f, t, r, h) = \sum_{b^{dep} \in \mathcal{B}^{dep}} \sum_{b^{tr} \in \mathcal{B}^{tr}} \left[ \overbrace{\sum_{x \in (b^{dep} \cup b^{tr})} \|x - r(f(x), t(x))\|_p^p}^{\mathcal{L}_{\text{recon}}} \right]$$
$$+ \underbrace{\lambda_{\text{match}} \left[ \log h\big(\{\text{sg}[f(x)] \mid x \in b^{dep}\}\big) - \log h\big(\{f(x) \mid x \in b^{tr}\}\big) \right]}_{\mathcal{L}_{\text{match}}}, \tag{9}$$

where $\mathcal{L}_{\text{recon}}$ denotes the reconstruction loss defined by the $p$-norm ($p = 1$ and $p = 2$ yielding MAE and MSE, respectively), $\mathcal{L}_{\text{match}}$ denotes the adversarial loss, $\lambda_{\text{match}} \in \mathbb{R}_*^+$ is a pre-factor controlling the trade-off between the loss terms, and $\text{sg}[\cdot]$ denotes the "stop-gradient" operator that behaves as the identity function but with zero partial derivatives. The overall objective, encompassing $f$, $t$, $r$, and $h$ can then be formulated in terms of $\mathcal{L}_{\text{enc}}$ as:

$$\min_{f,t,r} \max_{h} \mathcal{L}_{\text{enc}}(f, t, r, h) . \tag{10}$$

This equation is computed over batches of bags and the discriminator is trained to map a bag of samples from the training set and the deployment set to a binary label: 1 if the bag is adjudged to have been sampled from the deployment set, 0 if from the training set. For the discriminator to be able to classify sets of samples, it needs to be permutation-invariant along the bag dimension – that is, its predictions should take into account dependencies between samples in a bag while being invariant to the order in which they appear. For aggregating information over the bags, we use a self-attention-based (Vaswani et al., 2017) pooling layer in which the query vector is averaged over the bag dimension. For more details see Appendix G.3. Furthermore, in Sec. 4.3 (and Appendix H.1), we validate that having the discriminator operate over sets (bags) of samples rather than independent samples (with the same balancing scheme) is essential for achieving good and robust (w.r.t. balancing quality) performance.

Our goal is to disentangle $x$ into two subspaces: a subspace $z$, representing the class, and a subspace $\tilde{a}$, representing the attribute. For the problem to be well-posed, it is essential that bags differ only in terms of which groups are present. We thus sample the bags according to the following set of rules which operationalise $\Pi$ (see Fig. 9 in Appendix G.1 for a visualisation of the effect of these rules):

1. Bags of the deployment set are sampled so as to be approximately balanced with respect to $a$ and $y$ (all combinations of $a$ and $y$ should appear in equal number).
2. For bags from the training set, all possible values of $y$ should appear with equal frequency. Without this constraint, there is the risk of $y$ being encoded in $\tilde{a}$.
3. Bags of the training set should furthermore exhibit equal representation of each attribute within classes so long as rule 2 is not violated. For classes that do not have complete $\mathcal{A}$-support, the missing combinations of $(a, y)$ need to be substituted with a sample from the same class – i.e., if $a \notin \mathcal{A}^{tr}(y)$ we instead sample $a'$ randomly from a uniform distribution over $\mathcal{A}^{tr}(y)$.

### 3.3 Perfect bags

Borrowing from the FairML literature (Chouldechova, 2017; Kleinberg et al., 2016), we refer to a bag in which all elements of $\mathcal{G}$ appear in equal proportions as a "perfect bag" (even if the balancing is only approximate). Our pipeline can be broken down into two steps: 1) sample perfect bags from an unlabelled deployment set; and 2) produce disentangled representations using the perfect bags via adversarial support-matching as described in Sec. 3.2. A visual overview of our pipeline is given in Fig. 9 in Appendix G.1 while pseudocode is given in Algorithm 1.

**Constructing perfect bags via clustering.** One may use any off-the-shelf semi-supervised clustering algorithm, $C$, to uncover the $|\mathcal{G}|$ group-correspondent clusters required for perfect-bag construction. The

---

**Algorithm 1** Adversarial Support Matching

---

**Input:** Number of encoder updates $N^{\text{enc}}$, number of discriminator updates $N^{\text{disc}}$, encoders $f$ and $t$, decoder $r$, discriminator $h$, training set $\mathcal{D}^{tr}$, deployment set $\mathcal{D}^{dep}$
**Output:** Debiaser $f$ with learned invariance to $A$

   **for** $i \leftarrow 1$ to $N^{\text{enc}}$ **do**                                                         ▷ Encoder update loop
      Sample batches of perfect bags $\mathcal{B}^{tr} \sim \mathcal{D}^{tr}$ and $\mathcal{B}^{dep} \sim \mathcal{D}^{dep}$ using $\Pi$ (Eq. 4)
      Compute $\mathcal{L}^{\text{enc}}$ using Eq. 10
      Update $f$, $t$, and $r$ by descending in the direction $\nabla \mathcal{L}^{\text{enc}}$
      **for** $j \leftarrow 1$ to $N^{\text{disc}}$ **do**                                   ▷ Discriminator update loop
         Sample batches of perfect bags $\mathcal{B}^{tr} \sim \mathcal{D}^{tr}$ and $\mathcal{B}^{dep} \sim \mathcal{D}^{dep}$ using $\Pi$ (Eq. 4)
         Compute $\mathcal{L}^{\text{match}}$ using Eq. 10
         Update $h$ by ascending in the direction $\nabla \mathcal{L}^{\text{match}}$
      **end for**
   **end for**

---

avenue of developing better clustering pipelines is an orthogonal one that we leave to other works to pursue; the goal of this work is to establish a framework for mitigating the formalised AB problem with the clusters assumed given. For the Coloured MNIST and CelebA experiments featured in Sec. 4.1 and Sec. 4.2, respectively, we employ the ranking-statistics-based clustering method of Han et al. (2020) for its simplicity and strong empirical performance; for the NICO++ experiments we sidestep this choice and instead simulate clustering of specific levels of accuracy by injecting noise into the ground-truth labels with inter-cluster transitions governed by centroid-similarity within a CLIP (Radford et al., 2021) encoder's embedding space – full details of the procedure can be found in Appendix J.

As a result of clustering, the data points in the deployment set $\mathcal{D}^{dep}$ are labelled with cluster assignments generated by clustering algorithm, $C$, giving $\mathcal{D}^{dep}_C = \{(x_i, c_i)\}$, $c_i = C(z_i)$, so that we can form perfect bags from $\mathcal{D}^{dep}_C$ by sampling all clusters at equal rates; there is no need for application of the $\Pi$ operator since the deployment set is complete w.r.t. $\mathcal{G}$. We note that we do *not* have to associate the clusters with specific $a$ or $y$ labels as the labels are not directly used for supervision.

Balancing bags based on clusters instead of the true labels introduces an error, which we can try to bound. For this error-bounding, we assume that the probability distribution distance measure used in Eq. 5 is the *total variation distance* ($TV$). The proof can be found in Appendix E.

**Proposition 2.** *If $q_f(Z)$ is a data distribution on $\mathcal{Z}$ that is a mixture of $n_y \cdot n_a$ Gaussians, which correspond to all the unique combinations of $y \in \mathcal{Y}$ and $a \in \mathcal{A}$, and $p_f(Z)$ is any data distribution on $\mathcal{Z}$, then without knowing $y$ and $a$ on $q_f$, it is possible to estimate*

$$\sum_{a \in \mathcal{A}} \sum_{y \in \mathcal{Y}} TV(p_f|_{A \in \Pi(a,y), Y=y}, q_f|_{A=a, Y=y}) \tag{11}$$

*with an error that is bounded by $\tilde{O}(\sqrt{1/N})$ with high probability, where $N$ is the number of samples drawn from $q_f$ for learning.*

## 4 Experiments

We perform experiments on three publicly available image datasets – Coloured MNIST (following a similar data-generation process to Kehrenberg et al., 2020), CelebA (Liu et al., 2015), and NICO++ (Zhang et al., 2023) (following the split-construction procedure of Yang et al., 2023) – and one tabular dataset, American Community Survey (ACS). We report the `Robust Accuracy` – the minimum accuracy over all attributes – as the primary metric for Coloured MNIST and CelebA. For NICO++, we additionally report `Robust TPR` – analogously, the minimum TPR over all attributes – and designate this as the primary metric for consistency with prior work (Yang et al., 2023). Additional plots showing the Accuracy, Positive Rate, TPR, and TNR ratios can be found in Appendix G.6.

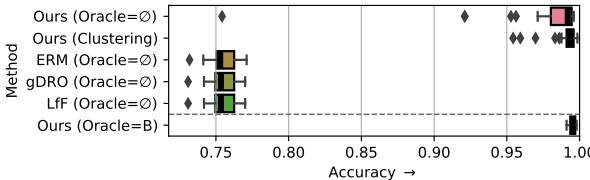 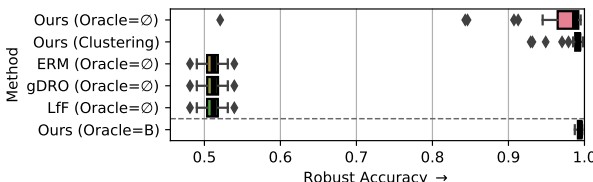

Figure 2: Results for Coloured MNIST from **30 replicates** for the AB scenario in which purple fours constitute the namesake missing group. The clustering accuracy for `Ours (Oracle=∅)` was $96\% \pm 6\%$. Our method consistently outperforms the baselines, which fare no better than random for the attribute with the missing group. Per expectation, the median and IQR of our method are positively and negatively correlated, respectively, with how well the bags of the deployment set are balanced, with `Ours (Oracle=B)` providing an upper bound for this. Indeed, in one case `Ours (Oracle=∅)` underperforms the baselines, but through clustering the `Robust Accuracy` (the minimum accuracy computed over the subgroups) is kept above 95%.

For methodological nomenclature, we use `Oracle=·` to denote oracle (ground-truth) labelling of the deployment set w.r.t. its argument $\cdot \in \{B, Y, A, \varnothing\}$. Parametrisation with B denotes bag (or batch) balancing using the ground-truth groups (made available only to the sampler); Y and A the furnishing of ground-truth targets and attributes, respectively for supervision, or strictly model-selection if the parameters are asterisked (Y*, A*); $\varnothing$ the absence of an oracle yet included for notational consistency. Furthermore, we use '&' to denote the composition of oracles, such that `Oracle=B&Y` should be interpreted as the oracle that balances bags using the ground-truth groups and additionally makes available the ground-truth targets. This overall scheme allows us to succinctly convey the amount of information required by each method. We compare the performance of our supporting-matching method when coupled with three different bag-balancing methods: 1) with clustering, via ranking statistics (`Ours (Clustering)`), or via simulated clustering in the case of NICO++ (in the context of a sensitivity analysis); 2) without balancing, that is, with the deployment set $\mathcal{D}^{dep}$ being used as is (`Ours (Oracle=∅)`); 3) with balancing using the ground-truth attributes and class labels (`Ours (Oracle=B)`) that are practically unobservable but afford an upper bound.

### 4.1 Coloured MNIST

Appendix F provides a description of the dataset and the settings used for $D^{dep}$ and $D^{tr}$. Each group is then a combination of digit-class (class label) and colour (attribute). We begin by considering a binary, 2-digit/2-colour, variant of the dataset with $\mathcal{Y} = \{2, 4\}$ and $\mathcal{A} = \{\text{green}, \text{purple}\}$. (Appendix D.1 provides results for 3-digit/3-colour variant.) To simulate the AB (attribute bias) setting, we set $\mathcal{A}^{tr}_{Y=4} = \{\text{green}\}$.

To establish the effectiveness of our method, we compare it against three baselines. The first is `ERM (Oracle=∅)`, a classifier trained with cross-entropy loss on this data; the second is `gDRO (Oracle=∅)` (Sagawa et al., 2019), which minimises the worst-case training loss over predefined attributes but is only applicable when $|\mathcal{A}^{tr}| > 1$; the third is `LfF (Oracle=∅)` (Nam et al., 2020) which reweights the cross-entropy loss using the predictions of a purposely-biased sister network. For fair comparison, the training set is balanced according to the rules defined in Sec. 3 for all baselines.

Fig. 2 shows the results for the AB setting. We see that the performance of our method directly correlates with how balanced the bags are, with the ranking of the different balancing methods being `Oracle=B` > `Clustering` > `Oracle=∅`. Even without balancing, our method significantly outperforms the baselines.

### 4.2 CelebA

To demonstrate our method generalises to real-world computer vision problems, we consider the CelebA dataset Liu et al. (2015) comprising over 200,000 images of different celebrities. The dataset comes with per-image annotations of physical and affective attributes such as smiling, gender, hair colour, and age. Since the dataset exhibits a natural imbalance with respect to $\mathcal{G}$, we perform no additional sub-sampling of either the training set or the deployment set. We predict "smiling" as the class label and use the binary

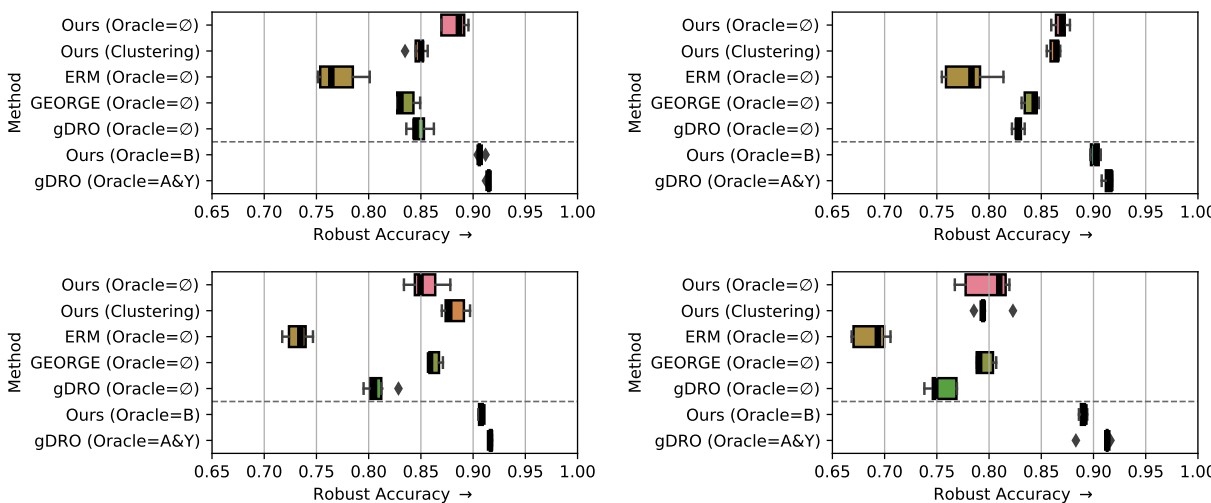

Figure 3: Results obtained from **five replicates** for the CelebA dataset. The task is to predict "smiling" vs "non-smiling" with the attribute being *gender*. The four groups are dropped in a leave-one-out fashion from the training set (**Top Left**: smiling females; **Top Right**: smiling males; **Bottom Left**: non-smiling females; **Bottom Right**: non-smiling males), while the deployment set is kept fixed. `Robust Accuracy` refers to the minimum accuracy computed over the attribute partitions. Our method consistently performs on par with or outperforms `GEORGE (Oracle=∅)` (which in turn outperforms `ERM (Oracle=∅)`. We note that in some of the runs, `GEORGE (Oracle=∅` performed no better than random – these results were truncated for visibility but can be found in Fig. 7. Given *indirect* supervision from the deployment set in the form of oracle-balancing, our method performs similarly to `gDRO (Oracle=A&Y)` that receives *direct* supervision.

label, "gender", as the attribute. Here, we consider the AB setting but rather than just designating one missing group, we repeat our experiments with each group being dropped in turn. As before, we evaluate our method under three balancing schemes and compare with ERM (`ERM (Oracle=∅)`) and gDRO trained on only the labelled training data (`gDRO (Oracle=∅)`). We also compare with two other variants of gDRO: 1) `gDRO (Oracle=A&Y)`, a variant that is trained with access to the ground-truth labels of the deployment set, thus providing an upper-bound on the downstream classification performance; 2) `GEORGE (Oracle=∅` (Sohoni et al., 2020), which follows a two-step procedure of first clustering to obtain the labels for hidden attributes, and then using these labels to train a robust classifier using gDRO. Sohoni et al. (2020) consider a different version of the problem (termed *hidden-stratification*) in which the class labels are known for all samples but the attribute labels are missing entirely. We adapt `GEORGE (Oracle=∅)` to our setting by modifying the ranking-statistics clustering algorithm to predict $P(Y|X)$ and $P(A|X)$, instead of the joint distribution $P(A, Y|X)$, allowing us to propagate the class labels from $D^{tr}$ to $D^{dep}$ (see Appendix I).

Fig. 3 shows the results for experiments for each missing groups, showing similar trends across all instantiations of the AB scenario. `gDRO (Oracle=A&Y)` consistently achieves the best performance according to both metrics, with `Ours (Oracle=B)` consistently coming in second. We note that while both methods use some form of oracle, `Oracle=A&Y` provides *all* class and attribute labels to its algorithm, whereas `Oracle=B` only balances the bags. Despite the large difference in the level of supervision, the margin between the two oracle methods is slim. We observe that clustering in many cases impairs performance which can be explained by poor clustering of the missing group (∼60% accuracy). CelebA exhibits a natural imbalance with respect to gender/smiling albeit not a significant one, such that random sampling represents a reasonable perfect-bag approximation. Nonetheless, among the non-oracle methods, variants of our method consistently match or outperform the baselines. While the plots show `GEORGE` can perform strongly in this AB scenario, we note that for several of the missing groups, the method failed catastrophically in one of the five replicates. We have removed the data points here in the interest of visibility; the complete versions of the plots can be found in Appendix D.2. The fact that `GEORGE (Oracle=∅)` leverages both the training and deployment sets

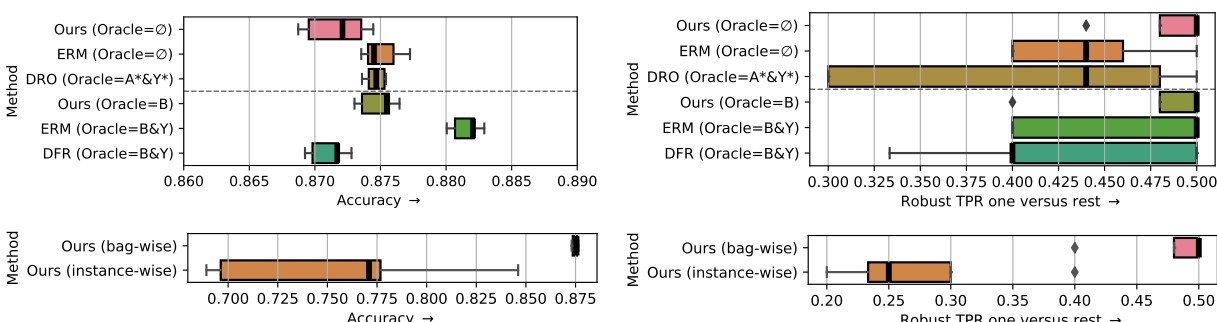

Figure 4: Results, in the form of box plots, for the NICO++ dataset constructed according to Yang et al. (2023) to simulate the AB scenario. These results represent the aggregation over five replicates, each replicate associated with a different PRNG seed for model initialisation but with the dataset fixed across all replicates for comparability with Yang et al. (2023). We report results in terms of both (aggregate) `Accuracy` (**left**) and `Robust TPR one versus rest` (**right**), with the latter serving as the primary metric (again, for comparability). Diamond markers denote outlier results. **Top**: Comparison with baselines. Our method, with (`Ours (Oracle=B)`) and without (`Ours (Oracle=∅)`) balancing, outperforms the baselines in terms of both medians and IQRs based on `TPR one versus rest`; all methods perform similarly based on `Accuracy`. **Bottom**: Ablation results for the NICO++ dataset with the type of discriminator (bag-wise versus instance-wise) being the ablated factor. Shown are the results for the bag-wise discriminator (`Ours (bag-wise)`, per the top plot, and an instance-wise discriminator (`Ours (instance-wise)`) with an identical architecture save for the removal of the aggregator, such that its losses are computed for each sample independently.

in a semi-supervised way with clustering makes it the baseline most comparable to our method. However, its performance is much more dependent on the clustering step.

## 4.3 NICO++

To demonstrate the scalability to cases where both classes and attributes are non-binary and the resulting number of groups, $|\mathcal{G}|$, is large, we turn to the recently-introduced NICO++ dataset (Zhang et al., 2023), with splits created in accordance with Yang et al. (2023) where the dataset is used to realise the *attribute generalisation* scenario analogous to our AB one. The data is taken from Track 1 of Zhang et al. (2023), with samples spanning 60 classes and 6 attributes, giving rise to 360 groups, marking a considerable upscaling of the problem compared with the datasets dealt with previously; the classes represent different object categories while the attributes correspond to different environmental factors (specifically, $\mathcal{A} = \{\text{autumn}, \text{dim}, \text{grass}, \text{outdoor}, \text{rock}, \text{water}\}$), disjointly. Groups with fewer than 75 samples are designated as the missing groups and removed from the training set; 50 and 25 samples from each group are set aside for the deployment and test sets, respectively, where we have adapted the scheme to our setup by co-opting the original validation set as the deployment set. Since the memory complexity of the exact-bag-construction scales linearly in $|\mathcal{G}|$, we resort to an approximate bag-balancing algorithm that amortises the cost by balancing in expectation. Namely, to construct each bag, for $split \in \{tr, dep\}$, we uniformly sample $N^{\tilde{A}} \leq |\mathcal{A}^{split}|$ attributes for each class $y'$ (*with* replacement if $\mathcal{A}^{split}_{Y=y'} \subsetneq \mathcal{A}$), and then sample a data point uniformly from each of the resulting groups, such that we have bags of size $N^{\tilde{A}} \cdot |\mathcal{Y}|$.

For the baselines, we complement the before-featured gDRO and ERM (both oracle and non-oracle variants) baselines with in DFR (Kirichenko et al., 2022) and DRO (Hashimoto et al., 2018), both using oracles to differing degrees. We straightforwardly adapt DFR by using the deployment set, with target and bag oracles, for the entailed last-layer retraining (`DFR (Oracle=B&Y)`). While DRO (`DRO (Oracle=A*&Y*)`) does not require such direct supervision, it is sensitive to the choice of the dual variable, $\eta$; we afford the baseline the advantage of having this variable selected to maximise deployment-set performance.

Remaining consistent with Yang et al. (2023), we evaluate the performance of all methods primarily in terms of the `Robust TPR one versus rest` (this is equivalent to computing the minimum of the group-

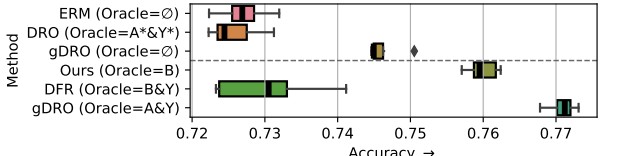 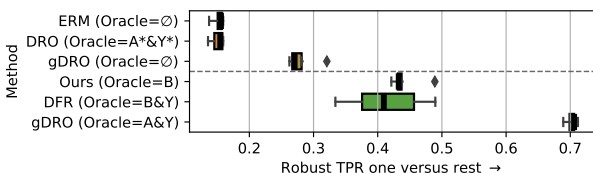

Figure 5: Results, in the form of box plots, for the ACSEmployment dataset (Ding et al., 2021) modified per 4.4 to simulate the AB scenario; these results represent the aggregation over five replicates, each replicate associated with separate PRNG seeds for model initialisation and dataset splitting. Shown are both the results in terms of (aggregate) `Accuracy` (**left**) and `Robust TPR one versus rest` (**right**), the latter to be considered as the primary metric for this dataset. Diamond markers denote outlier results.

wise accuracy scores and thus is commonly referred to as worst-group accuracy in the DRO literature; Sagawa et al., 2019). In Fig. 4 (**top**), we observe that, according to said metric, our method consistently outperforms the baselines, irrespective of whether the bags are balanced, exhibiting both higher medians and narrower IQRs. All methods perform similarly along the `Accuracy` axis, noting the scale of the abscissa. While theory dictates that our method improves proportionally with the balance of the bags, we surmise that the stochasticity of the approximate bag-sampler offsets this; nonetheless, this observation speaks to the robustness of our method, despite its being predicated on the idea of exact bag balance emulative of the supports of the training and deployment distributions. To expand on this observation, we conduct a sensitivity analysis w.r.t. the noise of the deployment set's balancing labels. The resultant plots, along with the noise-simulation procedure are available in Appendix J, but the main takeaway reinforces what is seen here, that the noise level does not have a significant impact on the performance of the model – if anything its effect is salutary suggesting it may serve as a source of implicit regularisation as characterised for stochastic gradient descent (SGD) generally (Sekhari et al., 2021).

Finally, we investigate the importance of casting the problem as a support-matching one, that is, having, *ceteris paribus*, the discriminator discriminate at the bag level rather than the instance level, the latter being the norm for adversarial domain adaptation methods (Ganin et al., 2016). Namely, we rerun `Ours (Oracle=B)` from Fig. 4 (**bottom**) (relabelled as `Ours (bag-wise)`) with only this aspect of the discriminator ablated (yielding `Ours (instance-wise)`); the comparison plots are given in Fig. 4 (**bottom**). With this ablation, our method fails along both metric axes for the majority of the five replicates, falling far short of any of the baselines; though the unablated and ablated methods may share a theoretical optimum, this equivalence does not appear to hold empirically, suggesting that bag-discrimination may engender better-behaved training dynamics.

## 4.4  American Community Survey (Tabular)

To gauge the performance of our proposed algorithm on tabular data, we turn to an instantiation of the American Community Survey (ACS) dataset introduced in Ding et al. (2021), consisting of data derived from US Census surveys collected over the years 2014-2018. Specifically, we tailor to our problem setting the *ACSEmployment* variant in which the task is to predict whether an individual is employed (*employment*), thus constituting a binary prediction task, $y \in \{0, 1\}$, and focus on the subset of the data collected from Florida in 2018. We designate *disability* (whether an individual declared (1), or did not declare (2), a disability) as the attribute, $a \in [2]$, and set $\mathcal{M}$ to $\{(0, 2)\}$, translating to the omission of employed individuals *with* a declared disability – the minority group at around 10% of the population – to simulate the AB problem for the training set. For more details on the dataset composition, see Appendix F.2. The results of the ACS experiments are graphed in Fig. 5, showing the bag-oracle version of our method, `Ours (Oracle=B)`, to outperform `DFR` in terms of the median `Robust TPR one versus rest`, despite not receiving direct supervision from the deployment set, and to be competitive with `gDRO (Oracle=A&Y)`, representative of a general upper-bound on performance, in terms of `Accuracy`.

## 5    Conclusion

In this paper, we formalised the problem of systematic bias or dataset curation resulting in one or more groups having zero labelled data; we hope our doing so stimulates serious consideration for it in the planning, building, and evaluating systems. This complements concurrent research by Yang et al. (2023) wherein the same problem (with different formalism) is alluded to, but its solution left as an open question. We proposed an approach based on constructing hierarchically-balanced bags from an unlabelled deployment set. We then match the support, instead of the raw distributions, of the training and deployment sets in representation space, to learn attribute-invariant representations, an outcome we prove corresponds to the optimum of the proposed objective function. We empirically validate our framework on a range of datasets and instantiations of the *Attribute Bias* (AB) problem: Coloured MNIST (Kehrenberg et al., 2020) as a synthetic image dataset (with both binary and multiclass variants), CelebA (Liu et al., 2015) and NICO++ (Yang et al., 2023) as real-world image datasets (binary and multiclass, respectively), and ACS (Ding et al., 2021) as a tabular dataset showcasing generality. We empirically show that it is possible to maintain high performance for samples with classes with incomplete support in a robust fashion.

## Acknowledgments

This research was funded by the European Union. Views and opinions expressed are however those of the author(s) only and do not necessarily reflect those of the European Union or the European Health and Digital Executive Agency (HaDEA). Neither the European Union nor the granting authority can be held responsible for them. European Research Council under the European Union's Horizon 2020 research and innovation programme Grant Agreement no. 851538 - BayesianGDPR, Horizon Europe research and innovation programme Grant Agreement no. 101120763 - TANGO. Novi Quadrianto is also supported by BCAM Severo Ochoa accreditation CEX2021-001142-S/MICIN/AEI/10.13039/501100011033. Viktoriia Sharmanska is currently at Epic Games.

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

# A  Notation

Table 1: Notation and abbreviations used throughout the paper

| Notation/Abbreviation | Definition |
|---|---|
| $\mathcal{X}$ | The input domain. |
| $\mathcal{Y}$ | The domain of class labels. |
| $\mathcal{A}$ | The domain of attribute labels. |
| $\mathcal{G}$ | The set of (intersectional) groups, where a "group" refers to a unique pair (two-tuple) of $a$ and $y$. |
| $Y$ | Discrete random variable, defined on $\mathcal{Y}$, encoding the target class. |
| $y$ | A realisation of the random variable $Y$ |
| $A$ | Discrete random variable, defined on $\mathcal{A}$, encoding the attribute. |
| $a$ | A realisation of the random variable $A$ |
| $\mathcal{Z}$ | The (latent) attribute-invariant encoding space mapped to by encoder $f(\cdot)$. |
| $Z$ | Continuous random variable, defined on $\mathcal{Z}$. |
| $z$ | A realisation of the random variable $Z$. |
| $f$ | Encoder function mapping from the input domain $\mathcal{X}$ to a attribute-invariant embedding space $\mathcal{Z}$. |
| $c$ | Classification head mapping from the embedding space generated by $f$ to the probability simplex over $\mathcal{Y}$: $c : \mathcal{Z} \to \triangle^{|\mathcal{Y}|}$. |
| $\Gamma$ | Classification model composed of encoder $f$ and classification head $c$, $\Gamma \triangleq c \circ f$. |
| $\mathcal{D}^{tr}$ | Training set of size $N^{tr}$. |
| $\mathcal{D}^{te}$ | Test set of size $N^{te}$ on which we ultimately wish to deploy our model. |
| $\mathcal{D}^{dep}$ | Deployment set of size $N^{dep}$ that is unlabelled w.r.t. both $a$ and $y$ but has support over the unobserved labels matching that of the test set. |
| $\mathcal{A}^{tr}_{Y=y'}$ | The set of observed attributes within class $y'$ in the training set, $\mathcal{D}^{tr}$. |
| $h$ | Discriminator network trained to discriminate bags between samples drawn from the training set from bags of samples drawn from the deployment set. |
| $\mathcal{I}(\cdot ; \cdot)$ | Mutual Information |
| SC | Spurious Correlations; correlations arising between the target and a secondary attribute that is acausal w.r.t. said target but that models can exploit to learn shortcut solutions that lead to low loss on the training set but poor performance at deployment time when the correlations no longer hold. |
| $\mathcal{M}$ | The set of missing groups, $\mathcal{G}^{te} \setminus \mathcal{G}^{te}$. |
| AB | *Attribute Bias* setting characterised by certain class-attribute combinations missing from the training set yet with all classes and attributes represented, marginally. This is a form of sampling bias that induces classifiers to learn attribute-specific shortcuts that fail to generalise to the group-complete test set. |

# B  Related work

**Invariant learning**  Sohoni et al. (2020) and Creager et al. (2021) both consider a similar problem, where the data also exhibits a two-level hierarchy formed by classes and attributes (the latter being called "subgroups" in those works). In contrast to our work, however, there is no additional bias in the data; while they may be unobserved, the labelled data is assumed to have complete class-conditional support over the attributes. As such, these methods are not directly applicable to the particular form of the problem we consider. Like us, Sohoni et al. (2020) uses semi-supervised clustering to uncover the hidden attributes,

however their particular clustering method requires access to the class labels not afforded by the deployment set, as does the training of the robust classifier.

**Unsupervised domain adaptation**  In unsupervised domain adaptation (UDA), there are typically one or more source domains, for which training labels are available, and one or more unlabelled target domains to which we hope to generalise the classifier. A popular approach for solving this problem is to learn a representation that is invariant to the domain using adversarial networks (Ganin et al., 2016) or non-parametric discrepancy measures such as MMD (Gretton et al., 2012).

There are two ways in which one can compare UDA to our setting: 1) by treating the attribute partitions as domains; and 2) by treating the training and the deployment set as "source" and "target" domains, respectively. The first comparison is exploited in algorithm fairness, yet does not carry over to our setting in which the labelled data contains *incomplete* domains. When all groups from a given domain are missing then there are no domains to be matched, and even when this is not the case, matching will result in misalignment due to differences in class-conditional support. The second comparison is more germane but ignores an important aspect of our problem: the presence of spurious correlations.

Similar to us, Tong et al. (2022) utilise adversarial methods to align the support of two distributions in a semi-supervised regime – specifically, they propose to use symmetric support difference as a divergence measure which they realise using a discriminator. However, their method focuses on label-shift in the UDA setting and does not consider the hierarchical structure that exists within the source (training) and target (deployment) domains, and as such they do not consider the notion of "missing groups" that can arise due to said structure – the characterisation of this problem is one of the two main contributions of this work (the other being our proposed solution). Furthermore, the discriminator used therein is applied instance-wise; we show in section H that allowing the discriminator to model inter-sample relationships has tangible additional benefits when performing support-matching.

**Multiple instance learning**  Multiple instance learning (Maron & Lozano-Pérez, 1998) is a form of weakly-supervised learning in which samples are not labelled individually as part of a set or *bag* of samples. In the simplest (binary) case, a bag is labelled as positive if there is a single instance of a positive class contained within it, and negative otherwise. In our case, we can view the missing groups as constituting the positive classes, which leads to all bags (a term we use throughout the paper distinctly from "batches") from the deployment set being labelled as positive, and all bags from the training set being labelled as negative. Given this labelling scheme, we make use of an adversarial set-classifier to align the supports of the training and deployment sets in the representation space of an encoder network.

**Positive unlabelled learning**  Learning from positive and unlabelled data, or *PU learning*, refers to the binary-classification setup in which the labelled training data consists of only positive samples while additional unlabelled data is assumed to contain both positive and negative samples (Liu et al., 2002; 2003; Bekker & Davis, 2020). This is analogous to our problem setting if we consider the positive class to be all samples from groups represented in the training set, collectively, while the missing groups collectively make up the negative class. However, the goal here is not merely to learn the classification boundary between the present and missing groups but to learn to classify the target class of a given sample independently of its attribute. This is equivalent to requiring that a classifier trained to distinguish between positive and negative classes, according to the aforementioned PU learning setup, from the pre-logits layer of our desired classifier, be maximally entropic. We propose to use adversarial learning to achieve this.

**Hierarchical classification**  Hierarchical classification (Silla & Freitas, 2011) refers to tasks where a (pre-defined) hierarchy of classes exists – the top of the hierarchy corresponds to coarse labels and the bottom to very fine-grained labels – and the goal is to robustly assign classes to instances such that the fine-grained class assignments can be wrong but the coarse-grained class assignments are almost always correct. A canonical example of a hierarchy of classes can be found in taxonomy, wherein each organism (taxon) is classified according to several taxonomic ranks, the bottom-most ranks being 'genus' and 'species'. Even if the classifier should incorrectly classify the species, it should correctly classify the genus. Though our

problem setup does have a hierarchical element, w.r.t. how the bias is imparted, the prediction task defined on it is not itself hierarchical, such that hierarchical classification is not of direct relevance to our work.

## C Results in tabular form

We also present the results in tabular form. The results from Figure 2 can be found in Table 2. For the results from Figure 3 see tables 3, 4, 5, and 6. And for Figure 4, see Table 7.

Table 2: Results for Colored MNIST with 2 groups missing. Same results as in Figure 2.

| Method | Acc. ↑ | Rob. Acc. ↑ | PR ratio | TPR ratio | TNR ratio |
|---|---|---|---|---|---|
| Ours (Oracle=∅) | $0.98 \pm 0.05$ | $0.95 \pm 0.09$ | $0.89 \pm 0.19$ | $0.91 \pm 0.19$ | $0.995 \pm 0.005$ |
| Ours (Clustering) | $0.99 \pm 0.01$ | $0.985 \pm 0.017$ | $0.95 \pm 0.04$ | $0.986 \pm 0.013$ | $0.99 \pm 0.03$ |
| ERM (Oracle=∅) | $0.755 \pm 0.009$ | $0.511 \pm 0.012$ | $0.0 \pm 0.0$ | $0.0 \pm 0.0$ | $0.994 \pm 0.004$ |
| gDRO (Oracle=∅) | $0.755 \pm 0.009$ | $0.512 \pm 0.012$ | $0.001 \pm 0.005$ | $0.001 \pm 0.005$ | $0.994 \pm 0.004$ |
| LfF (Oracle=∅) | $0.755 \pm 0.009$ | $0.511 \pm 0.012$ | $0.0 \pm 0.0$ | $0.0 \pm 0.0$ | $0.994 \pm 0.004$ |
| Ours (Oracle=B) | $0.995 \pm 0.002$ | $0.994 \pm 0.003$ | $0.96 \pm 0.03$ | $0.99 \pm 0.006$ | $0.997 \pm 0.003$ |

Table 3: Missing group: smiling females. Same results as in Figure 3.

| Method | Acc. ↑ | Rob. Acc. ↑ | PR ratio | TPR ratio | TNR ratio |
|---|---|---|---|---|---|
| Ours (Oracle=∅) | $0.89 \pm 0.006$ | $0.883 \pm 0.011$ | $0.93 \pm 0.03$ | $0.9 \pm 0.03$ | $0.928 \pm 0.007$ |
| Ours (Clustering) | $0.87 \pm 0.004$ | $0.848 \pm 0.007$ | $0.957 \pm 0.01$ | $0.85 \pm 0.02$ | $0.943 \pm 0.005$ |
| ERM (Oracle=∅) | $0.824 \pm 0.011$ | $0.771 \pm 0.019$ | $0.8 \pm 0.02$ | $0.67 \pm 0.03$ | $0.925 \pm 0.014$ |
| GEORGE (Oracle=∅) | $0.79 \pm 0.14$ | $0.76 \pm 0.15$ | $0.95 \pm 0.02$ | $0.88 \pm 0.017$ | $0.95 \pm 0.03$ |
| gDRO (Oracle=∅) | $0.87 \pm 0.005$ | $0.848 \pm 0.009$ | $0.989 \pm 0.006$ | $0.838 \pm 0.014$ | $0.934 \pm 0.006$ |
| Ours (Oracle=B) | $0.912 \pm 0.003$ | $0.907 \pm 0.003$ | $0.834 \pm 0.01$ | $0.983 \pm 0.006$ | $0.951 \pm 0.006$ |
| gDRO (Oracle=A&Y) | $0.921 \pm 0.002$ | $0.915 \pm 0.001$ | $0.771 \pm 0.019$ | $0.975 \pm 0.01$ | $0.987 \pm 0.008$ |

Table 4: Missing group: smiling males. Same results as in Figure 3.

| Method | Acc. ↑ | Rob. Acc. ↑ | PR ratio | TPR ratio | TNR ratio |
|---|---|---|---|---|---|
| Ours (Oracle=∅) | $0.897 \pm 0.004$ | $0.869 \pm 0.006$ | $0.541 \pm 0.015$ | $0.757 \pm 0.018$ | $0.919 \pm 0.003$ |
| Ours (Clustering) | $0.893 \pm 0.003$ | $0.863 \pm 0.005$ | $0.572 \pm 0.013$ | $0.773 \pm 0.015$ | $0.956 \pm 0.005$ |
| ERM (Oracle=∅) | $0.857 \pm 0.006$ | $0.78 \pm 0.02$ | $0.36 \pm 0.04$ | $0.51 \pm 0.06$ | $0.9 \pm 0.03$ |
| GEORGE (Oracle=∅) | $0.873 \pm 0.003$ | $0.84 \pm 0.007$ | $0.58 \pm 0.03$ | $0.76 \pm 0.03$ | $0.966 \pm 0.003$ |
| gDRO (Oracle=∅) | $0.88 \pm 0.002$ | $0.828 \pm 0.004$ | $0.477 \pm 0.01$ | $0.656 \pm 0.011$ | $0.935 \pm 0.013$ |
| Ours (Oracle=B) | $0.915 \pm 0.002$ | $0.902 \pm 0.004$ | $0.66 \pm 0.03$ | $0.88 \pm 0.02$ | $0.963 \pm 0.011$ |
| gDRO (Oracle=A&Y) | $0.924 \pm 0.002$ | $0.914 \pm 0.004$ | $0.71 \pm 0.02$ | $0.927 \pm 0.017$ | $0.988 \pm 0.008$ |

### C.1 Limitation and intended use

Although having zero labelled examples for some groups is not uncommon due to the effects of systematic bias or dataset curation, we should make a value-judgement on the efficacy of the dataset with respect to a task. We can then decide whether or not to take corrective action as described in this paper. A limitation of the presented approach is that, for constructing the perfect bags used to train the disentangling algorithm, we have relied on knowing the number of clusters *a priori*, something that, in practice, is perhaps not the case. However, for person-related data, such information can, for example, be gleaned from recent census data. (See also Appendix H.2 for results with misspecified numbers of clusters.) One difficulty with automatic determination of the number of clusters is the need to ensure that the small clusters are correctly identified.

Table 5: Missing group: non-smiling females. Same results as in Figure 3.

| Method | Acc. ↑ | Rob. Acc. ↑ | PR ratio | TPR ratio | TNR ratio |
|---|---|---|---|---|---|
| Ours (Oracle=∅) | $0.872 \pm 0.009$ | $0.854 \pm 0.016$ | $0.58 \pm 0.02$ | $0.865 \pm 0.015$ | $0.76 \pm 0.04$ |
| Ours (Clustering) | $0.888 \pm 0.007$ | $0.882 \pm 0.01$ | $0.663 \pm 0.009$ | $0.913 \pm 0.007$ | $0.86 \pm 0.02$ |
| ERM (Oracle=∅) | $0.802 \pm 0.006$ | $0.732 \pm 0.011$ | $0.502 \pm 0.007$ | $0.88 \pm 0.005$ | $0.47 \pm 0.03$ |
| GEORGE (Oracle=∅) | $0.8 \pm 0.14$ | $0.78 \pm 0.16$ | $0.68 \pm 0.02$ | $0.911 \pm 0.017$ | $0.9 \pm 0.05$ |
| gDRO (Oracle=∅) | $0.847 \pm 0.007$ | $0.808 \pm 0.011$ | $0.558 \pm 0.015$ | $0.885 \pm 0.014$ | $0.66 \pm 0.03$ |
| Ours (Oracle=B) | $0.913 \pm 0.002$ | $0.908 \pm 0.002$ | $0.694 \pm 0.012$ | $0.921 \pm 0.006$ | $0.94 \pm 0.015$ |
| gDRO (Oracle=A&Y) | $0.923 \pm 0.001$ | $0.917 \pm 0.001$ | $0.729 \pm 0.017$ | $0.948 \pm 0.011$ | $0.981 \pm 0.013$ |

Table 6: Missing group: non-smiling males. Same results as in Figure 3.

| Method | Acc. ↑ | Rob. Acc. ↑ | PR ratio | TPR ratio | TNR ratio |
|---|---|---|---|---|---|
| Ours (Oracle=∅) | $0.867 \pm 0.008$ | $0.8 \pm 0.02$ | $0.91 \pm 0.04$ | $0.944 \pm 0.004$ | $0.74 \pm 0.04$ |
| Ours (Clustering) | $0.866 \pm 0.005$ | $0.798 \pm 0.013$ | $0.93 \pm 0.03$ | $0.953 \pm 0.007$ | $0.75 \pm 0.03$ |
| ERM (Oracle=∅) | $0.821 \pm 0.007$ | $0.687 \pm 0.015$ | $0.755 \pm 0.018$ | $0.93 \pm 0.004$ | $0.53 \pm 0.03$ |
| GEORGE (Oracle=∅) | $0.79 \pm 0.14$ | $0.73 \pm 0.14$ | $0.947 \pm 0.007$ | $0.954 \pm 0.005$ | $0.82 \pm 0.09$ |
| gDRO (Oracle=∅) | $0.849 \pm 0.006$ | $0.754 \pm 0.013$ | $0.86 \pm 0.02$ | $0.952 \pm 0.011$ | $0.67 \pm 0.02$ |
| Ours (Oracle=B) | $0.91 \pm 0.001$ | $0.89 \pm 0.003$ | $0.864 \pm 0.017$ | $0.987 \pm 0.006$ | $0.93 \pm 0.012$ |
| gDRO (Oracle=A&Y) | $0.917 \pm 0.009$ | $0.908 \pm 0.012$ | $0.791 \pm 0.014$ | $0.985 \pm 0.006$ | $0.986 \pm 0.007$ |

Table 7: Results for NICO++. Same results as in Figure 4.

| Method | Acc. ↑ | Rob. TPR OvR ↑ |
|---|---|---|
| Ours (Oracle=∅) | $0.872 \pm 0.002$ | $0.48 \pm 0.02$ |
| ERM (Oracle=∅) | $0.875 \pm 0.001$ | $0.44 \pm 0.04$ |
| DRO (Oracle=A*&Y*) | $0.875 \pm 0.001$ | $0.4 \pm 0.09$ |
| Ours (Oracle=B) | $0.875 \pm 0.001$ | $0.48 \pm 0.04$ |
| ERM (Oracle=B&Y) | $0.882 \pm 0.001$ | $0.46 \pm 0.05$ |
| DFR (Oracle=B&Y) | $0.871 \pm 0.001$ | $0.43 \pm 0.06$ |

A cluster formed by an underrepresented attribute can be easily overlooked by a clustering algorithm in favour of larger but less meaningful clusters.

# D    Additional experiments

## D.1    Results for 3-digit 3-colour variant of Coloured MNIST

To investigate how our method scales with the number of groups, we look to a 3-digit, 3-colour variant of the dataset in the *attribute bias* setting where four groups are missing from $\mathcal{D}^{tr}$. Results for this configuration are shown in fig. 6. We see that the performance of `Ours (Oracle=∅)` approaches that of `Ours (Oracle=B)`. We suspect this is because balancing is less critical with the increased number of groups strengthening the training signal. As inter-attribute ratios do not make for suitable metric for non-binary $A$, we instead quantify the invariance of the predictions to the attribute with the HGR maximal correlation (Rényi, 1959).

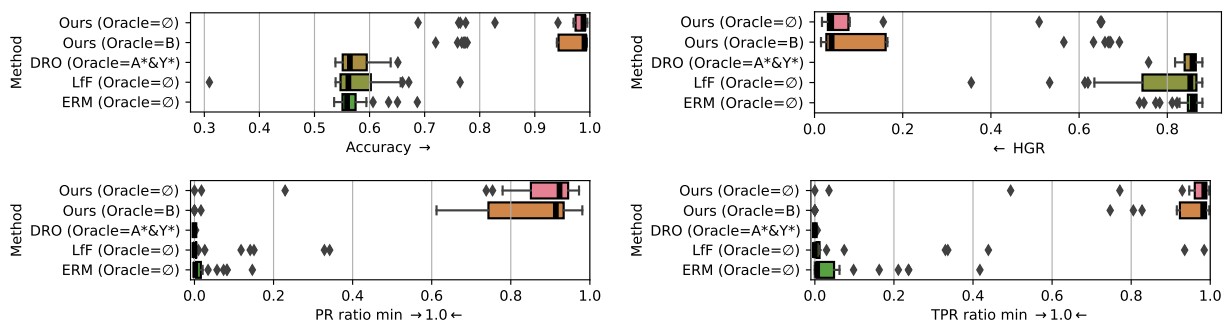

Figure 6: Results from **30 replicates** for the Coloured MNIST dataset with three digits: '2', '4' and '6'. Four combinations of digit and colour are missing: green 2's, blue 2's, blue 4's and green 6's. **Left**: Accuracy. **Right**: Hirschfeld-Gebelein-Rényi maximal correlation Rényi (1959) between $A$ and $Y$.

## D.2    Extended Results for CelebA

As alluded to in main text, for three out of four of the missing gender/smiling quadrants, the `GEORGE` baseline produced an extreme outlier for one out of the five total replicates - these outliers were omitted from the plots to ensure the discriminability of the other results. We reproduce the full, untruncated versions of these plots here in fig. 7. We have also included `Accuracy` metric in fig. 7.

# E    Theoretical analysis

In this section, we present our theoretical results concerning the validity of our support-matching objective and the bound on the error introduced into it by clustering. We use notation consistent with that used throughout the main text.

## E.1    Sampling function for the objective

The stated objective uses the following helper function:

$$\Pi(a,y) = \begin{cases} \{a\} & \text{if } \mathcal{A}^{tr}_{Y=y} = \mathcal{A} \\ \mathcal{A}^{tr}_{Y=y} & \text{otherwise} . \end{cases} \tag{12}$$

This helper function determines which $a$ value in the training set an $a$-$y$ pair from the deployment set is mapped to. (The $y$ value always stays *the same* when mapping from deployment set to training set.) To demonstrate the usage of this function, we consider the example of binary Coloured MNIST with $\mathcal{A} =$

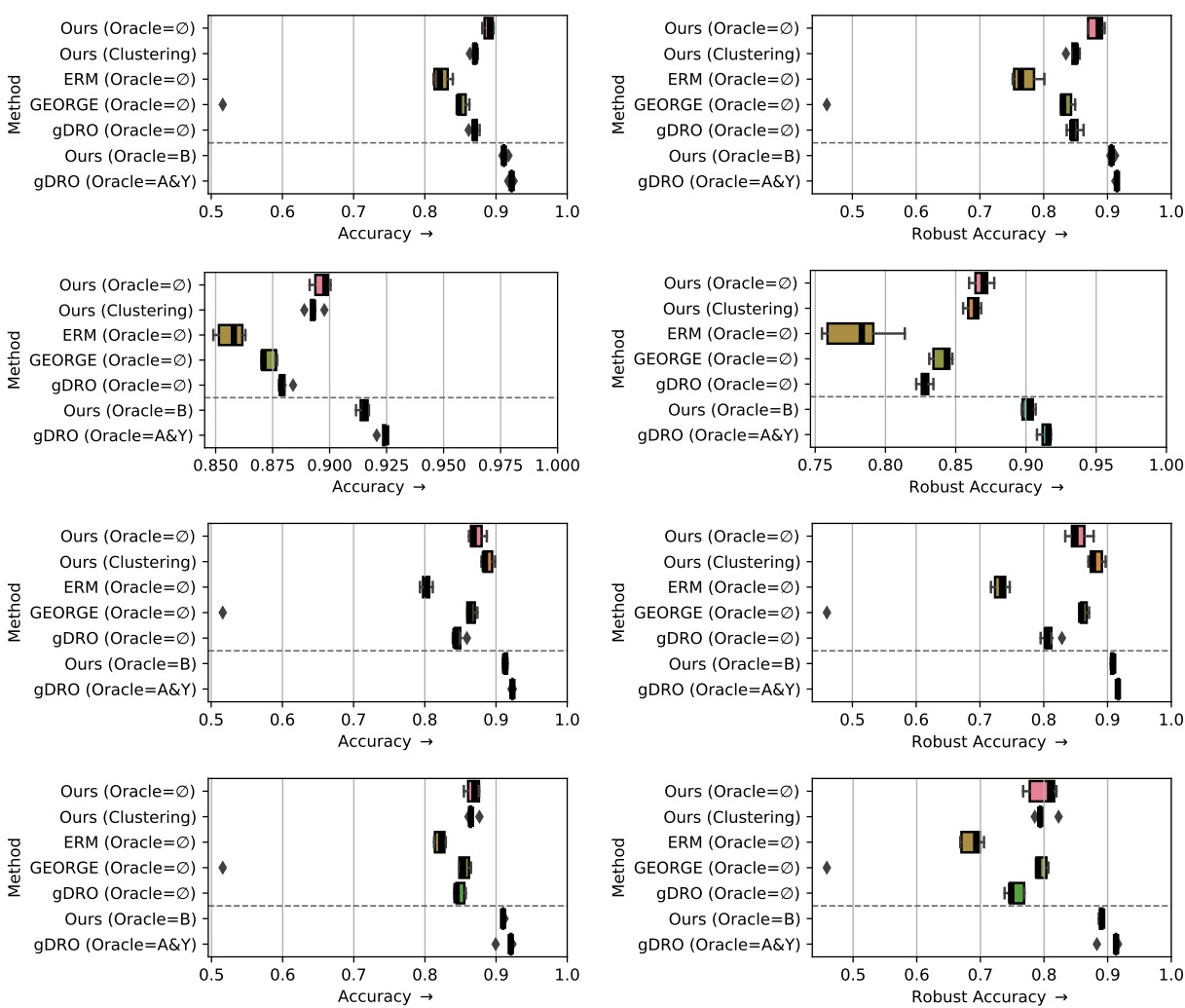

Figure 7: Results from **5 replicates** for the CelebA dataset for the *attribute bias* scenario. The task is to predict "smiling" vs "non-smiling" while the attribute is *gender*. The four groups are dropped one at a time from the training set (**first row**: smiling females; **second row**: smiling males; **third row**: non-smiling females, **fourth row**: non-smiling males), while the deployment set is kept fixed. "Robust Accuracy" refers to the minimum accuracy computed over the attributes.



(a) Correct matching procedure.  (b) *In*correct matching procedure.

Figure 8: The two natural matching procedures for one missing group in the training set. Only figure 8a (left) produces the desired invariance.

{purple, green} and $\mathcal{Y} = \{2, 4\}$ where the training set is missing $(a = \text{purple}, y = 4)$. In this case, $\Pi$ takes on the following values:

$$\Pi(\text{purple}, 2) = \{\text{purple}\} \tag{13}$$

$$\Pi(\text{green}, 2) = \{\text{green}\} \tag{14}$$

$$\Pi(\text{purple}, 4) = \mathcal{A}^{tr}_{y=4} = \{\text{green}\} \tag{15}$$

$$\Pi(\text{green}, 4) = \mathcal{A}^{tr}_{y=4} = \{\text{green}\} \tag{16}$$

It is essential that $(a = \text{purple}, y = 4)$ from the deployment set is mapped to $(a = \text{green}, y = 4)$ from the training set, and not $(a = \text{purple}, y = 2)$. This procedure is illustrated in fig. 8a, and contrasted with an incorrect procedure based on balancing the bag according to $a$ in 8b – such a procedure would result in invariance to $y$ instead of $a$, which is obviously undesirable.

In practice, we use the following sampling function $\pi$ to implement $\Pi$, sampling from it for all $(a, y) \in \mathcal{A} \times \mathcal{Y}$:

$$\pi(a, y) = \begin{cases} x \sim P^{tr}(x|A = a, Y = y), & \text{if } \mathcal{A}^{tr}_{Y=y} = \mathcal{A} \\ x \sim P^{tr}(x|A = a', Y = y), a' \sim \text{uniform}(A^{tr}), & \text{otherwise .} \end{cases} \tag{17}$$

With the assumption that our data follows a two-level hierarchy and all digits appear in the training set, the above sampling function $\pi$ traverses the first level which corresponds to the class-level information, and *samples* the second level which corresponds to attribute-level information when we have missing groups.

If this procedure results in bags that are too large for the used hardware (as was the case for us w.r.t. NICO++, given that $|G| = 360$), the procedure can be approximated further by dropping the bag-wise constraints laid out in Sec. 5. The following describes, for the training set, the case of $N^{\tilde{A}} = 1$ (1 sample per attribute per bag), which results in bags of size $|\mathcal{Y}|$:

$$\tilde{\pi}(y) = x \sim P^{tr}(x|A = a, Y = y), a \sim \text{uniform}(A^{tr}_{Y=y}) . \tag{18}$$

Here, we simply sample an attribute $a$ from those that are present for $y$ and then pick a sample from the group defined by $(a, y)$.

### E.2 Implication of the objective

We restate proposition 1 and present the proof.

We prove here that an encoding $f$ satisfying the objective is invariant to $a$, at least in those cases where the class does not have full $a$-support (which is exactly the case where it matters).

**Proposition 3.** *If $f$ is such that*

$$p_f|_{A \in \Pi(a,y), Y=y} = q_f|_{A=a, Y=y} \quad \forall a \in \mathcal{A}, y \in \mathcal{Y} \tag{19}$$

and $P^{tr}$ and $P^{dep}$ are data distributions that correspond to the real data distribution $P$, except that some $a$-$y$-combinations are less prevalent, or, in the case of $P^{tr}$, missing entirely, then, for every $y' \in \mathcal{Y}$, there is either full coverage of $a$ for $y'$ in the training set ($\mathcal{A}^{tr}_{Y=y'} = \mathcal{A}$), or the following holds:

$$P(A = a'|f(X) = z', Y = y') = \frac{1}{n_a} \ . \tag{20}$$

In other words: for $Y = y'$, $f(x)$ is not predictive of $a$.

*Proof.* If $y'$ has full coverage of $\mathcal{A}$ in the training set, there is nothing to prove. So, assume $y'$ does not have full $\mathcal{A}$-support. That means $\Pi(a', y') = \mathcal{A}^{tr}_{Y=y'}$ for all $a' \in \mathcal{A}$. And so

$$P^{tr}(f(X) = z'|A \in \mathcal{A}^{tr}_{Y=y'}, Y = y') = P^{dep}(f(X) = z'|A = a', Y = y') \qquad \forall a' \in \mathcal{A} \tag{21}$$

The left-hand side of this equation does not depend on $a'$ and so the right-hand side must have the same value for all $a' \in \mathcal{A}$, which implies:

$$P^{dep}(f(X) = z'|A = a', Y = y') = P^{dep}(f(X) = z'|Y = y') \tag{22}$$

Now, by assumption, the different data distributions *train* and *deployment* only differ from the "true" distribution by the prevalence of the different $a$-$y$-combinations, with the *deployment* data distribution having all combinations but potentially not in equal quantity. However, as we restrict ourselves to a certain combination $(A = a', Y = y')$ in the above equation, the equation also holds in the true data distribution:

$$P(f(X) = z'|A = a', Y = y') = P(f(X) = z'|Y = y') \tag{23}$$

Then, using Bayes' rule, we get

$$\begin{aligned} &P(A = a'|f(X) = z', Y = y') \\ &= \frac{P(f(X) = z'|A = a', Y = y')P(A = a'|Y = y')}{P(f(X) = z'|Y = y')} \\ &= P(A = a'|Y = y') \ . \end{aligned} \tag{24}$$

Finally, in the true data distribution, we have a uniform prior: $P(A = a'|Y = y') = (n_a)^{-1}$. This concludes the proof. $\qquad \square$

### E.3 Bound on error introduced by clustering

As previously stated, in practice, no labels are available for the deployment set. Instead, we identify the relevant groupings by clustering. Such clustering cannot be expected to be perfect. So, how will clustering affect the calculation of our objective?

**Proposition 4.** *If $q_f(Z)$ is a data distribution on $\mathcal{Z}$ that is a mixture of $n_y \cdot n_a$ Gaussians, which correspond to all unique combinations of $y \in \mathcal{Y}$ and $a \in \mathcal{A}$, and $p_f(Z)$ is any data distribution on $\mathcal{Z}$, then without knowing $y$ and $a$ on $q_f$, we can estimate*

$$\sum_{a \in \mathcal{A}} \sum_{y \in \mathcal{Y}} TV(p_f|_{A \in \Pi(a,y), Y=y}, q_f|_{A=a, Y=y}) \tag{25}$$

*with an error that is bounded by $\tilde{O}(\sqrt{1/N})$ with high probability, where $N$ is the number of samples drawn from $q_f$ for learning.*

*Proof.* First, we produce an estimate $\hat{q}_f$ of $q_f$ using the algorithm from Ashtiani et al. (2020), which gives us a mixture-of-Gaussian distribution of $n_y \cdot n_a$ components with $TV(q_f, \hat{q}_f) \leq \tilde{O}(\sqrt{1/N})$ with high probability, where $N$ is the number of data points used for learning the estimate. Then, by Lemma 3 from Sohoni et al.

(2020), *there exists* a mapping $i$ from the components $k$ of the Gaussian mixture $\hat{q}_f$ to the $a$-$y$-combinations in $q_f$ such that

$$TV(q_f(Z|A = a, Y = y), \hat{q}_f(Z|k = i(a, y))) \leq \tilde{O}\left(\frac{1}{\sqrt{N}}\right) . \tag{26}$$

To make the notation more manageable, we introduce the following shorthands:

$$p_f(a, y) := p_f(Z|A \in \Pi(a, y), Y = y) \tag{27}$$
$$q_f(a, y) := q_f(Z|A = a, Y = y) \tag{28}$$
$$\hat{q}_f(a, y) := \hat{q}_f(Z|k = i(a, y)) \tag{29}$$

Our goal is then to find an estimate for $TV(p_f(a, y), q_f(a, y))$ (because we don't have access to $q_f$). The estimate in question will be $TV(p_f(a, y), \hat{q}_f(a, y))$ and the error of the estimate is given by

$$|TV(p_f(a, y), q_f(a, y)) - TV(p_f(a, y), \hat{q}_f(a, y)| . \tag{30}$$

In order to bound this error, we use the triangle equation and the fact that equation 26 provides a bound:

$$TV(p_f(a, y), q_f(a, y) \leq TV(p_f(a, y), \hat{q}_f(a, y)) + TV(\hat{q}_f(a, y), q_f(a, y))$$
$$\leq TV(p_f(a, y), \hat{q}_f(a, y)) + \tilde{O}(1/\sqrt{N}) \tag{31}$$
$$\Rightarrow TV(p_f(a, y), q_f(a, y) - TV(p_f(a, y), \hat{q}_f(a, y)) \leq \tilde{O}(1/\sqrt{N}) \tag{32}$$

Using the fact that $TV$ is symmetric, we can use the same process to also derive

$$TV(p_f(a, y), \hat{q}_f(a, y) - TV(p_f(a, y), q_f(a, y)) \leq \tilde{O}(1/\sqrt{N}) . \tag{33}$$

Together, these imply

$$|TV(p_f(a, y), q_f(a, y)) - TV(p_f(a, y), \hat{q}_f(a, y)| \leq \tilde{O}(1/\sqrt{N}) . \tag{34}$$

Thus, for the whole sum over $a$ and $y$, the error is bounded by

$$\sum_{a \in \mathcal{A}} \sum_{y \in \mathcal{Y}} |TV(p_f(a, y), q_f(a, y)) - TV(p_f(a, y), \hat{q}_f(a, y)| \tag{35}$$

$$\leq \sum_{a \in \mathcal{A}} \sum_{y \in \mathcal{Y}} \tilde{O}(\sqrt{1/N}) \leq n_a n_y \max_{(a, y) \in \mathcal{A} \times \mathcal{Y}} \tilde{O}(\sqrt{1/N}) \tag{36}$$

which is equivalent to just $\tilde{O}(\sqrt{1/N})$, given the definition of $\tilde{O}$ (see Ashtiani et al., 2020). $\qquad\square$

## F  Dataset Construction

### F.1  Coloured MNIST and biasing parameters

The MNIST dataset LeCun et al. (1998) consists of 70,000 (60,000 designated for training, 10,000 for testing) images of gray-scale hand-written digits. We colour the digits following the procedure outlined in Kehrenberg et al. (2020), randomly assigning each sample one of ten distinct RGB colours. Each group is then a combination of digit-class (class label) and colour (attribute). We use no data-augmentation aside from symmetrically zero-padding the images to be of size 32x32.

To simulate a more real-world setup where the data, labelled or otherwise, is not naturally balanced, we bias the Coloured MNIST training and deployment sets by downsampling certain colour/digit combinations. The proportions of each such combination *retained* in the *attribute bias* scenario (in which we have one group missing from the training set) are enumerated in table 8. For the 3-digit-3-colour variant of the problem, no biasing is applied to either the deployment set or the training set (the missing combinations are specified in the caption accompanying figure 14); this variant was experimented with only under the attribute-bias setting.

Table 8: Biasing parameters for the training (left) and deployment (right) sets of Coloured MNIST in the *attribute bias* setting.

| Combination | Proportion retained | |
|---|---|---|
| | training set | deployment set |
| (Y = 2, A = purple) | 1.0 | 0.7 |
| (Y = 2, A = green) | 0.3 | 0.4 |
| (Y = 4, A = purple) | 0.0 | 0.2 |
| (Y = 4, A = green) | 1.0 | 1.0 |

### F.2 Dataset composition

For the composition of the different CelebA splits, see tables 9, 10, 11, and 12.

For statistics about our split of the ACS, see table 13.

Table 9: Statistics for CelebA with missing group: smiling females.

| Train | | | | Deployment | | | |
|---|---|---|---|---|---|---|---|
| Smiling | Gender | Number | Fraction | Smiling | Gender | Number | Fraction |
| No | Female | 21657 | 39% | No | Female | 21718 | 26.8% |
| No | Male | 20278 | 36.5% | No | Male | 20231 | 25% |
| Yes | Female | 0 | 0% | Yes | Female | 25603 | 31.6% |
| Yes | Male | 13565 | 24.4% | Yes | Male | 13488 | 16.6% |

Table 10: Statistics for CelebA with missing group: smiling males.

| Train | | | | Deployment | | | |
|---|---|---|---|---|---|---|---|
| Smiling | Gender | Number | Fraction | Smiling | Gender | Number | Fraction |
| No | Female | 21657 | 32.1% | No | Female | 21718 | 26.8% |
| No | Male | 20278 | 30.1% | No | Male | 20231 | 25% |
| Yes | Female | 25539 | 37.9% | Yes | Female | 25603 | 31.6% |
| Yes | Male | 0 | 0% | Yes | Male | 13488 | 16.6% |

## G  Pipeline and optimisation details

### G.1  Overview of the support-matching pipeline

We present in Fig. 9 a high-level, visual overview of our support-matching pipeline capturing both the hierarchical sampling and modelling aspects that characterise our proposed method.

### G.2  Overview of model architecture

We give a more detailed explanation of the model used in our method. Fig. 10 shows the core of our method: the debiaser, $f$, which produces bags of encodings, $z$ – on both the training and the deployment set – which are then fed to a discriminator that tries to identify the origin of the bags. The discriminator uses batch-wise attention in order to consider a bag as a whole, which allows cross-comparisons.

Table 11: Statistics for CelebA with missing group: non-smiling females.

| Train | | | | Deployment | | | |
|-------|--------|--------|----------|---------|--------|--------|----------|
| Smiling | Gender | Number | Fraction | Smiling | Gender | Number | Fraction |
| No | Female | 0 | 0% | No | Female | 21718 | 26.8% |
| No | Male | 20278 | 34.1% | No | Male | 20231 | 25% |
| Yes | Female | 25539 | 43% | Yes | Female | 25603 | 31.6% |
| Yes | Male | 13565 | 22.8% | Yes | Male | 13488 | 16.6% |

Table 12: Statistics for CelebA with missing group: non-smiling males.

| Train | | | | Deployment | | | |
|-------|--------|--------|----------|---------|--------|--------|----------|
| Smiling | Gender | Number | Fraction | Smiling | Gender | Number | Fraction |
| No | Female | 21657 | 35.6% | No | Female | 21718 | 26.8% |
| No | Male | 0 | 0% | No | Male | 20231 | 25% |
| Yes | Female | 25539 | 42% | Yes | Female | 25603 | 31.6% |
| Yes | Male | 13565 | 22.3% | Yes | Male | 13488 | 16.6% |

Table 13: Statistics for ACS dataset, Florida 2018, with employment as the target and disability as the attribute, for splitting random seed 0. Other random seeds give almost the same statistics.

| Train | | | | Deployment | | | |
|-------|----------------|--------|----------|----------|----------------|--------|----------|
| Employed | Has disability | Number | Fraction | Employed | Has disability | Number | Fraction |
| No | Yes | 16011 | 14.1% | No | Yes | 4515 | 12.6% |
| No | No | 39475 | 34.8% | No | No | 11384 | 31.7% |
| Yes | Yes | 0 | 0% | Yes | Yes | 3378 | 9.4% |
| Yes | No | 57994 | 51.1% | Yes | No | 16607 | 46.3% |

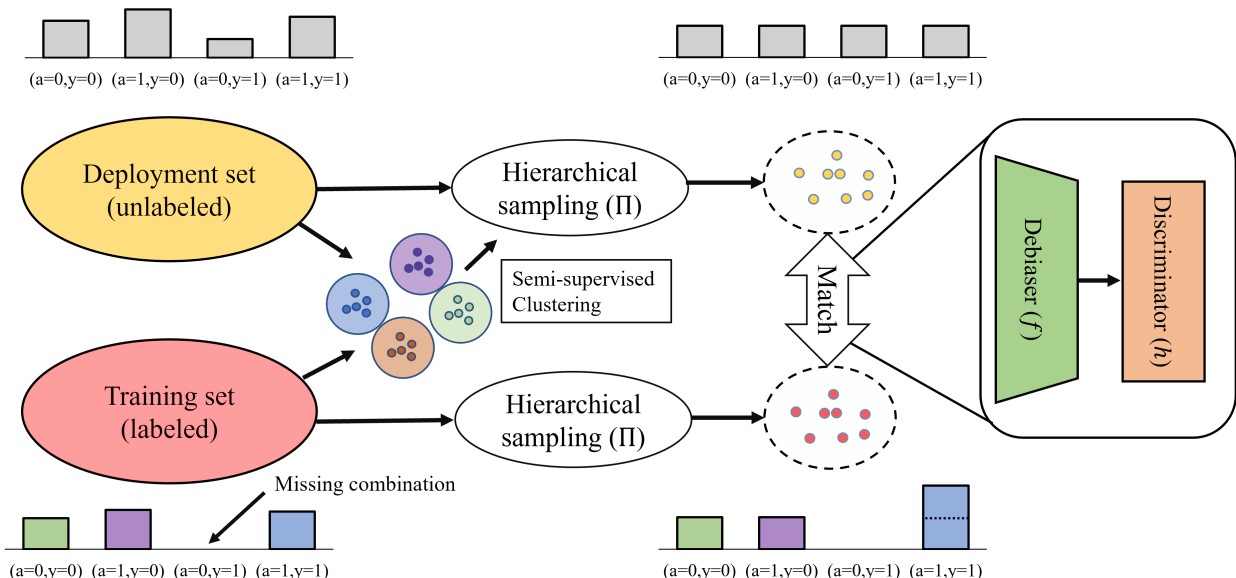

Figure 9: Visualisation of our support-matching pipeline. Bags are sampled from the training and deployment sets using the hierarchical sampling procedure described in Sec. 3 and defined functionally in Eq. 17. Since we cannot use ground-truth labels for hierarchical sampling of the deployment set, we use a semi-supervised clustering algorithm to produce balanced batches. In the event that certain combinations are missing, as shown here for $(a = 0, y = 1)$, the sampling on the training set substitutes the missing combinations with combinations that ensure equal representation of the target classes. The debiaser is adversarially trained to produce representations from which the source dataset cannot be reliably inferred by the discriminator. Assuming the bags are sufficiently balanced and $\mathcal{G}^{tr} \subsetneq \mathcal{G} = \mathcal{A} \times \mathcal{Y}$, the optimal debiaser is one that produces a representation $z$ that is invariant to $a$; we establish proof of this in Appendix E.2.

### G.3 Details of the attention mechanism

The *discriminator* function $h$ that predicts which dataset a bag of samples embedded in $z$ was sampled from should have the following property: $h((f(x)|x \in b)) = h((f(x)|x \in \tau(b)))$ for all permutations $\tau$, and all $f : \mathcal{X} \rightarrow \mathcal{Z}$. For the entirety of function $h$ – composed of sub-functions $h_1(h_2(h_3 \ldots)))$ – to have this property, it suffices that only the innermost sub-function, $\zeta$, does. While there are a number of choices when it comes to defining $\zeta$, we choose a weighted average $\zeta(b) = \frac{1}{|b|} \sum_{x \in b} \text{attention}(f(x), b) \cdot f(x)$, with weights computed according to a learned attention mechanism. The idea of using an attention mechanism for set-wise classification has been previously explored to great success by, e.g., Lee et al. (2019). We employ a bag-wise attention mechanism based on the scaled dot-product attention of Vaswani et al. (2017), where in our case we define $K$ and $V$ to be linear projections of $z$ ($zW_k$ and $zW_v$, respectively) and $Q$ to be the mean of another linear projection of $z$, $zW_q$, taken over the bag dimension.

$$\text{Attention}(Q, K, V) := \text{Softmax}\left(\frac{QK^T}{\sqrt{d}}\right)V$$

The output of $\zeta$ is then further processed by a series of fully-connected layers and the final output is the binary prediction for a given bag of samples.

### G.4 Training procedure and hyperparameters

The hyperparameters and architectures for the AutoEncoder (`AE`), Predictor and Discriminator subnetworks are detailed in Table 14 for the Coloured MNIST and CelebA datasets, and in Table 15 for the ACS dataset. The hyperparameters associated with our NICO++ experiments are tabulated separately in Table 16, owing

Table 14: Hyperparameters for Coloured MNIST and CelebA experiments.

| | **Coloured MNIST** 2-dig AB / 3-dig AB | **CelebA** |
|---|---|---|
| Input size | $3 \times 32 \times 32$ | $3 \times 64 \times 64$ |
| *AutoEncoder* | | |
| Levels | 4 | 5 |
| Level depth | 2 | 2 |
| Hidden units / level | $[32, 64, 128, 256]$ | $[32, 64, 128, 256, 512]$ |
| Activation | GELU | SiLU |
| Layer-wise Normalisation | - | LayerNorm |
| Downsampling op. | Strided Convs. | Strided Convs. |
| Reconstruction loss | MSE | MSE |
| Learning rate | $1 \times 10^{-3}$ | $1 \times 10^{-3}$ |
| *Clustering* | | |
| Batch size | 256 | 256 |
| AE pre-training epochs | 150 | 10 |
| Clustering epochs | 100 | 20 |
| Self-supervised loss | Cosine + BCE | Cosine |
| U (for ranking statistics) | 5 | 8 |
| *Support-Matching* | | |
| Batch size | 1/14 | 32 |
| Bag size | 256/18 | 8 |
| Training iterations | 8k/20k | 2k |
| Encoding ($z$) size[2] | 128 | 128 |
| Binarised $\tilde{s}$ | ✗ / ✓ | ✗ |
| $y$-predictor weight ($\lambda_1$) | 1 | 1 |
| $a$-predictor weight ($\lambda_2$) | 1 | 1 |
| Adversarial weight ($\lambda_3$) | $1 \times 10^{-3}$ | 1 |
| *Predictors* | | |
| Learning rate | $3 \times 10^{-4}$ | $1 \times 10^{-3}$ |
| *Discriminator* | | |
| Attention mechanism[3] | Gated | Gated |
| Hidden units pre-aggregation | $[256, 256]$ | $[256, 256]$ |
| Hidden units post-aggregation | $[256, 256]$ | $[256, 256]$ |
| Embedding dim (for attention) | 32 | 128 |
| Activation | GELU | GELU |
| Learning rate | $3 \times 10^{-4}$ | $1 \times 10^{-3}$ |
| Updates / AE update | 1 | 1 |

[1] Cross-entropy is used for categorical features, MSE for continuous features.

[2] Denotes the combined size of $\tilde{a}$ and $z$, with the former occupying $\lceil \log_2(\mathcal{A}) \rceil$ dimensions, the latter the remaining dimensions.

[3] The attention mechanism used for computing the sample-weights within a bag. *Gated* refers to gated attention proposed by Ilse et al. (2018), while *SDP* refers to the scaled dot-product attention proposed by Vaswani et al. (2017).

Table 15: Hyperparameters for ACS experiments.

| AutoEncoder | |
|---|---|
| Levels | 3 |
| Hidden units / level | $[95, 95, 95]$ |
| Activation | GELU |
| Layer-wise Normalisation | LayerNorm |
| Reconstruction loss | MSE |
| Learning rate | $1 \times 10^{-4}$ |
| **Support-Matching** | |
| Batch size | 32 |
| Bag size | 4 |
| Training iterations | 10k |
| Encoding ($z$) size[1] | 64 |
| Binarised $\tilde{s}$ | ✗ |
| $y$-predictor weight ($\lambda_1$) | 1 |
| $a$-predictor weight ($\lambda_2$) | 0 |
| Adversarial weight ($\lambda_3$) | $3 \times 10^{-2}$ |
| **Predictors** | |
| Learning rate | $10^{-4}$ |
| **Discriminator** | |
| Attention mechanism[2] | SDP |
| Embedding dim (for attention) | 512 |
| Activation | GELU |
| Learning rate | $10^{-4}$ |
| Updates / AE update | 3 |

[1] Denotes the combined size of $\tilde{a}$ and $z$, with the former occupying $\lceil \log_2(\mathcal{A}) \rceil$ dimensions, the latter the remaining dimensions.

[2] The attention mechanism used for computing the sample-weights within a bag. *Gated* refers to gated attention proposed by Ilse et al. (2018), while *SDP* refers to the scaled dot-product attention proposed by Vaswani et al. (2017).

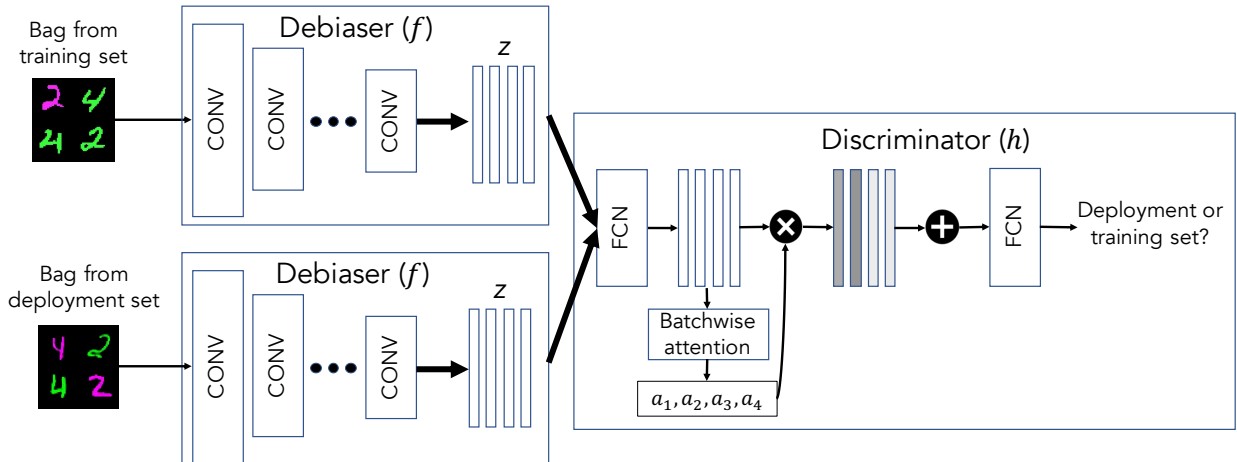

Figure 10: The main components of our support-matching algorithm, $f$ (debiaser) and $h$. The debiaser is trained to produce encodings, $z$ of the data that are invariant to the attribute differences. it. In order to determine whether a bag of encodings originates from the training set or the deployment set, the discriminator performs an attention-weighted aggregation over the bag dimension to model interdependencies between the samples, where $a_1, \ldots, a_4$ denote the attention weights assigned to each of the four samples making up said dimension. In the case of Coloured MNIST where purple fours constitute the missing group, the discriminator can identify an encoding of a bag from the training set by the absence of such samples as long as colour information is detectable in $z$. By learning an attribute-invariant representation, the debiaser conceals the origin of the bags (training versus deployment) from the discriminator.

Table 16: Hyperparameters for NICO++ experiments.

| Support-Matching | |
| --- | --- |
| Batch size | 1 |
| Bag size | 60 |
| Training iterations | 50k |
| Encoding ($z$) size[1] | 128 |
| Binarised $\tilde{s}$ | ✗ |
| $y$-predictor weight ($\lambda_1$) | 1 |
| $a$-predictor weight ($\lambda_2$) | 0 |
| Adversarial weight ($\lambda_3$) | $3 \times 10^{-2}$ |
| Predictors | |
| Learning rate | $10^{-5}$ |
| Discriminator | |
| Attention mechanism[2] | SDP |
| Embedding dim (for attention) | 512 |
| Activation | GELU |
| Learning rate | $10^{-5}$ |
| Updates / AE update | 5 |

[1] Denotes the combined size of $\tilde{a}$ and $z$, with the former occupying $\lceil \log_2(\mathcal{A}) \rceil$ dimensions, the latter the remaining dimensions.

[2] The attention mechanism used for computing the sample-weights within a bag. *Gated* refers to gated attention proposed by Ilse et al. (2018), while *SDP* refers to the scaled dot-product attention proposed by Vaswani et al. (2017).

to the fixed AE architecture – consisting of a ResNet50 (He et al., 2016) encoder and mirroring decoder – and lack of clustering. We train all models using the `AdamW` (Loshchilov & Hutter, 2018) optimiser.

For the Coloured MNIST and CelebA datasets, the baseline `ERM`, `DRO`, `LfF` (in the case of the former) and `gDRO` (in the case of the latter) models use a convolutional backbone consisting of one Conv-BN-LReLU block per "stage", with each stage followed by max-pooling operation to spatially downsample by a factor of two to produce the subsequent stage. This backbone consists of 4 and 5 stages for Coloured MNIST and CelebA, respectively. The output of the backbone is flattened and fed to a single fully-connected layer of size $|Y|$ in order to obtain the class-prediction, $\hat{y}_i$, for a given instance. To evaluate our method, we simply train a linear classifier on top of $z$; this is sufficient due to linear-separability being encouraged during training by the $y$-predictor. All baselines and downstream classifiers alike were trained for 60 epochs with a learning rate of $1 \times 10^{-3}$ and a batch size of 256.

Since, by design, we do not have labels for all groups the model will be tested on, and bias against these missing groups is what we aim to combat, properly validating, and thus conducting hyperparameter selection for models generally, is not straightforward. Indeed, performing model-selection for domain generalisation problems is well known to be a difficult problem Gulrajani & Lopez-Paz (2021). We can, however, use estimates of the mutual information between the learned-representation and $a$ and $y$ (which we wish to minimise w.r.t. to the former, maximise w.r.t. the latter) to guide the process, though optimising the model w.r.t. to these metrics obtained from only the training set does not guarantee generalisation to the missing groups. We can, however, additionally measure the entropy of the predictions on the encoded test set and seek to maximise it across all samples, or alternatively train a discriminator of the same kind used for distribution matching as a measure of the shift in the latent space between datasets. We use the latter approach (considering the combination of the learned distance between subspace distributions and reconstruction loss) to inform an extensive grid search over the hyperparameter space for our method.

For the `DRO` baseline, we allowed access to the labels of the test set for the purpose of hyperparameter selection, performing a grid search over multiple splits to avoid overfitting to any particular instantiation. Specifically, the threshold ($\eta$) parameter for `DRO` was determined by a grid search over the space $\{0.01, 0.1, 0.3, 1.0\}$.

### G.5   Visualisations of results

We show qualitative results of the disentangling for CelebA in Fig.11. With a successful disentangling, column 3 (visualisation of $z$) should show a version of the image that is "gender-neutral" (i.e., invariant to gender). Furthermore, column 4 (visualisation of $\tilde{a}$) should be invariant to the class label (i.e., "smiling"), so the images should either be all with smiles or all without smiles.

Fig. 12 shows attention maps for bags from the deployment set. We can see that the model pays special attention to those samples that are not included in the training set. For details, see the captions.

### G.6   Additional metrics

Figures 13 and 15 show the positive rate (PR) ratio, the true positive rate (TPR) ratio and the true negative rate (TNR) ratio as additional metrics for Coloured MNIST (2 digits) and CelebA. These are computed as the ratio of PR/TPR/TNR for samples with attribute $a = 0$ over the PR/TPR/TNR for samples with attribute $a = 1$; if this gives a number greater than 1, the inverse is taken. Similarly to the PR ratio reported in the main paper, these ratios give an indication of how much the prediction of the classifier depends on the attribute label $a$.

Figure 14 shows metrics specific to multivariate $a$ (i.e., non-binary $a$). We report the minimum (i.e. farthest away from 1) of the pairwise ratios (TPR/TNR ratio min) as well as the largest difference between the raw values (TPR/TNR diff max). Additionally, we compute the Hirschfeld-Gebelein-Rényi (HGR) maximal correlation Rényi (1959) between $A$ and $Y$, serving as a measure of dependence defined between two variables with arbitrary support.

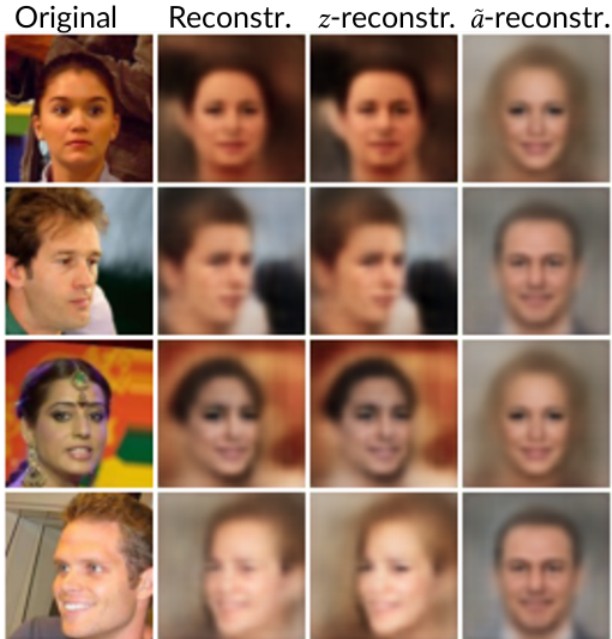

Figure 11: Visualisation of our method's solutions for the CelebA dataset, with "smiling females" as the missing group. Column 1 shows the original images from $x$ from the deployment set of CelebA. Column 2 shows plain reconstructions generated from $x_{recon} = g(f(x), t(x))$. Column 3 shows reconstruction with zeroed-out $\tilde{a}$: $g(f(x), 0)$, which effectively visualises $z$. Column 4 shows the result of an analogous process where $z$ was zeroed out instead.

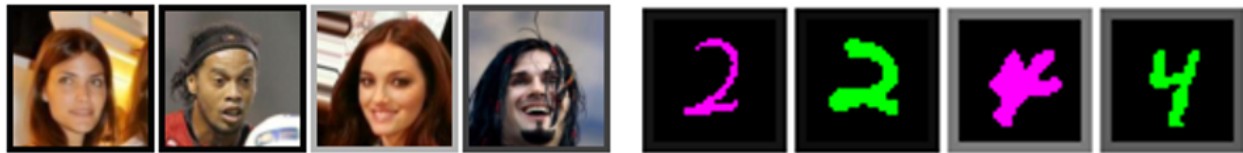

Figure 12: Example sample-wise attention maps for bags of CelebA (**left**) and Coloured MNIST (**right**) images sampled from a balanced deployment set. The training set is biased according to the *attribute bias* setting where for CelebA "smiling females" constitute the missing group and for Coloured MNIST purple fours constitute the missing group. The attention weights are used during the discriminator's aggregation step to compute a weighted sum over the bag. The attention weight assigned to each sample is proportional to the lightness of its frame, with black signifying a weight of 0, white a weight of 1. Those samples belonging to the missing group are assigned the highest weight as they signal from which dataset (training vs. deployment) the bag containing them was drawn from.

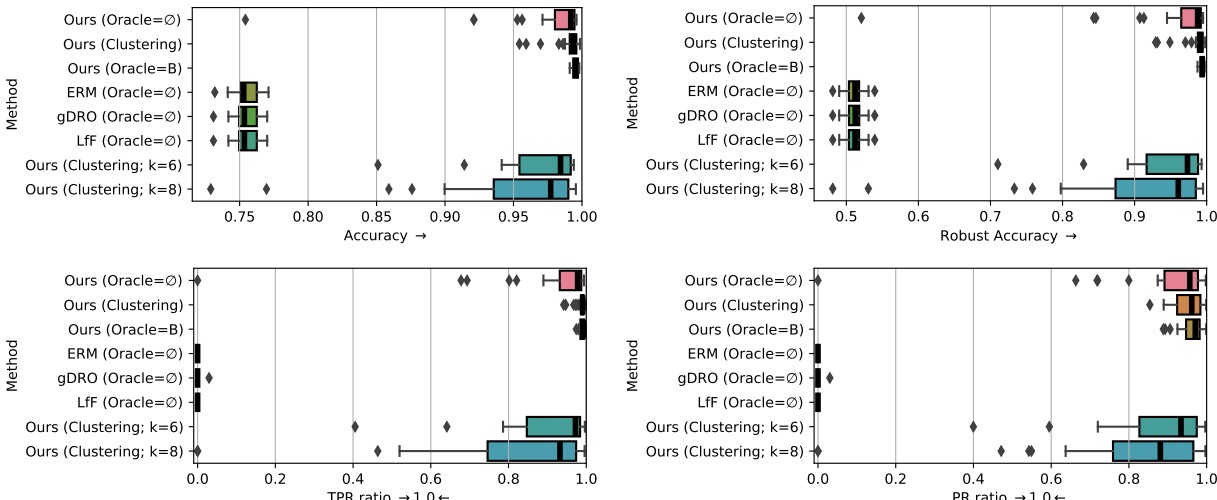

Figure 13: Results from **30 replicates** for the Coloured MNIST dataset with two digits, 2 and 4, with *attribute bias* for the colour 'purple': for purple, only the digit class '2' is present. **Top left**: Accuracy. **Top right**: Robust accuracy. **Bottom left**: True positive rate ratio. **Bottom right**: Positive rate ratio. For `Ours (Clustering)`, the clustering accuracy was $96\% \pm 6\%$. For an explanation of `Ours (Clustering; k=6/8)` see section H.2.

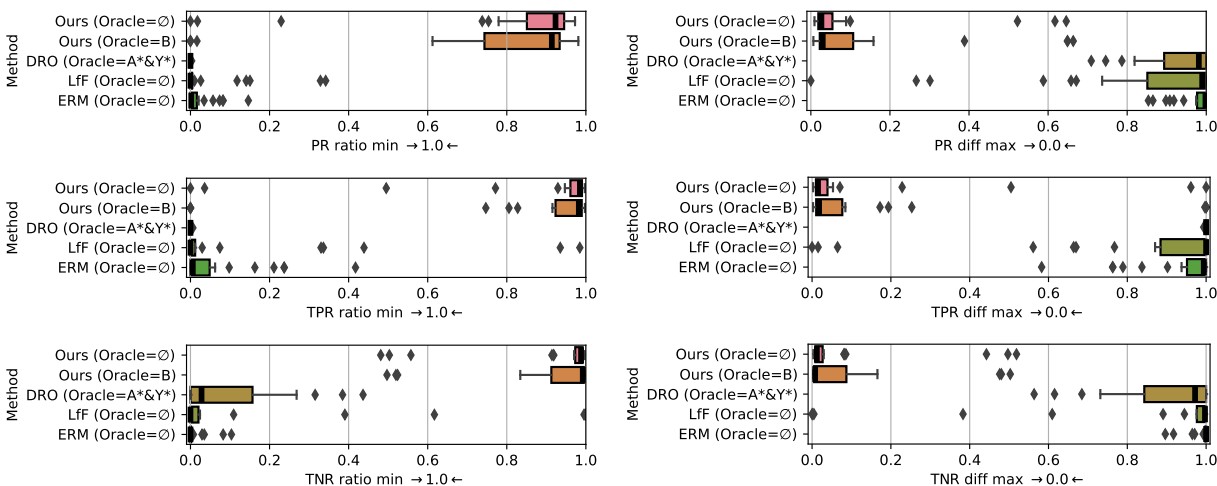

Figure 14: Results from **30 replicates** for the Coloured MNIST dataset with three digits: '2', '4' and '6'. Four combinations of digit and color are missing: green 2's, blue 2's, blue 4's and green 6's. **First row, left**: minimum of all positive rate ratios. **First row, right**: maximum of all positive rate differences. **Second row, left**: minimum of all true positive rate ratios. **Second row, right**: maximum of all true positive rate differences. **Third row, left**: minimum of all true negative rate ratios. **Third row, right**: maximum of all true negative rate differences.

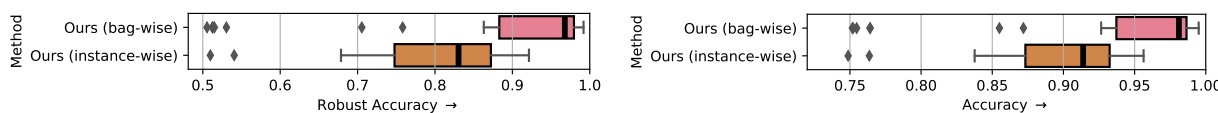

Figure 15: Plots of additional metrics for CelebA under the AB setting, where "smiling" is the class label and "gender" is the attribute label. These metrics are ratios computed between the *Male* and *Female* attribute with the largest of the two values involved always selected as the denominator. **Left:** TNR ratio. **Right**: TNR ratio.

# H    Ablation studies

## H.1    Using an instance-wise loss instead of a set-wise loss

Figure 16: Results from **30 replicates** comparing a *bag-wise* loss with an *instance-wise* loss for the Coloured MNIST dataset with two digits, 2 and 4, with *attribute bias* for the colour 'purple': for purple, only the digit class '2' is present. **Left**: Robust Accuracy. **Right**: Accuracy. Both types of runs are for the clustering setup. The clustering accuracy was $96\% \pm 6\%$.

See Fig. 16 for results on 2-digit Coloured MNIST under the *attribute bias* settings, for our method but with the loss computed instance-wise (as opposed to bag-wise) as is typical of adversarial unsupervised domain adaptation methods (e.g. Ganin et al. (2016)). All aspects of the method, other than those directly involved in the loss-computation, were kept constant – this includes the use of hierarchical balancing, despite the necessary removal of the aggregation layer meaning the discriminator is no longer sensitive to the bagging. It is clear that the aforementioned change to the loss drastically increases the variance (IQR) of the results and, at the same time, drastically reduces the median `Robust Accuracy` to the point of being only marginally above that of the `ERM (Oracle=∅` baseline, regardless of the chosen balancing scheme.

### H.2 Clustering with an incorrect number of clusters

We also investigate what happens when the number of clusters is set incorrectly. For 2-digit Coloured MNIST, we expect 4 clusters, corresponding to the 4 possible combinations of the binary class label $y$ and the binary attribute label $a$. However, there might be circumstances where the correct number of clusters is not known; how does the batch balancing work in this case? We run experiments with the number of clusters set to 6 and to 8, with all other aspects of the pipeline kept the same. It should be noted that this is a very naïve way of dealing with an unknown number of clusters. There are methods specifically designed for identifying the right number of clusters Hamerly & Elkan (2004); Chazal et al. (2013), and that is what would be used if this situation arose in practice.

The results can be found in Fig. 13. Bags and batches are constructed by drawing an equal number of samples from each cluster. Unsurprisingly, the method performs worse than with the correct number of clusters. When investigating how the clustering methods deal with the larger number of clusters, we found that it is predominantly those samples that do not appear in the training set which get spread out among the additional clusters. This is most likely due to the fact that the clustering is semi-supervised, with those clusters that occur in the training set having supervision. The overall effect is that the samples which are not appearing in the training set are over-represented in the drawn bags, which means it is easier for the adversary to identify where the bags came from, and the encoder cannot properly learn to produce an invariant encoding.

### H.3 Ablating other components of the method

In Fig. 17, we show results from removing the decoder function $r$ (referred to as "Without recon loss" in the plot). We also try removing the $y$-predictor – a classifier that predicts $y$ from the encoding $z$. The results show that both components improve "Robust TPR one versus rest", which is the main metric we care about.

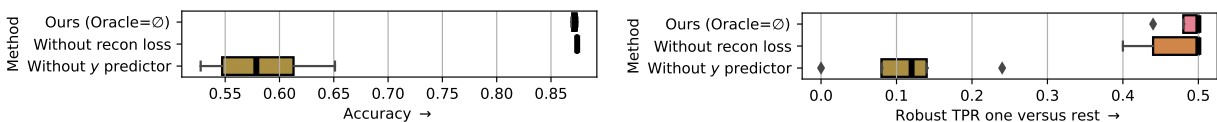

Figure 17: Results from ablating the reconstruction loss (i.e., decoder function $r$) and the $y$-predictor.

# I Adapting GEORGE

As discussed in the main text, GEORGE (Sohoni et al., 2020) was originally developed to address the uneven performance resulting from hidden stratification, though hidden stratification of a different kind to the one we consider. In Sohoni et al. (2020) the training set comes with (super-)class labels but without subclass (or *attribute* in our terminology) labels. The training set is unlabelled with respect to the subclass, but all superclass-subclass combinations (or "groups") are assumed to be present in the training data and therefore discoverable via clustering. (Note that the clustering in Sohoni et al. (2020) is – in contrast to our method – completely without supervision and there is nothing to guide the clustering towards discovering the attributes

of interest, apart from the assumption that they are the most salient.) On the other hand, in our setting, we do have access to all groups expected at deployment time, but not all of them are present in the training data – some are exclusively found in the *unlabelled* deployment set.

This necessitates propagating the labels from the training set to the deployment set, which can be done within the clustering step to ensure consistency between the cluster labels and the propagated superclass labels. Doing so requires us to modify the clustering algorithm such that instead of predicting each group independently of one another, we factorise the joint distribution of the super- and subclasses, $P(A, Y)$ into their respective marginal distributions, $P(Y)$ and $P(A)$. In practice, this is achieved by applying two separate cluster-prediction heads to the image representation, $z$: one, $\mu_y$, predicting the superclass, $y$, the other, $\mu_a$, predicting the subclass, $a$. This allows us to decouple the supervised loss for the two types of label and to always be able to recover $y$ due to having full supervision in terms of its ground-truth labels from the training set – this means we can identify all the $y$ clusters with the right $y$ labels.

With the outputs structured as just described, we can obtain the prediction for a given sample's group (which is needed to compute the unsupervised clustering loss and for balancing the deployment set), by taking the argmax of the vectorized outer product of the softmaxed outputs of the two heads:

$$\omega_i = \arg\max_k \ \text{vec}(\mu_y(z_i) \otimes \mu_a(z_i))_k \ , \tag{37}$$

$$k = 1, \ldots, |A \times Y|$$

where $\mu_y(z_i)$ and $\mu_a(z_i)$ are vectors, $\otimes$ is the outer product, and $\text{vec}(B)$ is the vectorisation of matrix $B$. After training the clustering model, we can then use it to generate predictions $\hat{Y}^{dep}$, as well as the cluster labels $\hat{\Omega}^{dep}$, for the deployment set, and use them together to train a robust classifier with gDRO Sagawa et al. (2019), as in Sohoni et al. (2020).

## J  Sensitivity Analysis

To assess the empirical robustness of our algorithm to noise in the bag-balancing, we conduct an sensitivity analysis using the NICO++ dataset. Using the same configuration used to collect the non-ablative results, we run our algorithm with the ground-truth labels used for balancing but with $\rho \in \{0\%, 20\%, \ldots, 60\%\}$ of the labels perturbed (flipped). Rather than randomly perturbing the labels by sampling from the set of complementary labels – i.e., $\tilde{g}_i \sim \text{uniform}(\mathcal{G} \setminus \{g_i\})$, with $g_i$ and $\tilde{g}_i$ denoting the ground-truth and the perturbed labels, respectively – which yields unrealistic perturbations due to the discounting the semantic relationships between groups (semantically similar groups are more likely to be confused by a clustering algorithm) we instead sample a perturbed label, $\tilde{g}_i$ from $\mathcal{G}$ with probability proportional to the similarity between the associated (featurised) sample and the $\tilde{g}_i$th centroid, $\phi_{\tilde{g}_i} \in \mathbb{R}^d$, with the constraint that the perturbed label does not equal the original one. This is done using features extracted by a pre-trained CLIP (Radford et al., 2021) visual encoder. Namely, the centroids $\Phi \in \mathbb{R}^{|\mathcal{G}| \times d}$ are computed as the group-conditional means of these features, over the deployment set, and their similarity with a given sample's features is measured using cosine similarity. Assuming $L_2$-normalised CLIP features, $\bar{z}_i^{\text{CLIP}} \in \mathbb{R}^d$, and prototypes, $\bar{\Phi}$, computed by their aggregation, the sampling scheme used to generate the perturbed label, $\tilde{g}_i$, for a given ground-truth label, $g_i$, can be written as

$$\tilde{g}_i \sim \text{Cat}(\mathcal{G}, \triangle_{\mathbf{w}}), \quad \triangle_{\mathbf{w}} \triangleq \frac{\mathbf{w}}{\sum_j \mathbf{w}_j}, \quad \mathbf{w} \triangleq \exp(\bar{z}_i^{\text{CLIP}} \cdot \bar{\Phi}\tau^{-1}) \odot (1 - e_{g_i}), \tag{38}$$

where Cat denotes the categorical distribution with support $\mathcal{G}$ and sampling probabilities $\triangle_w$, $\tau \in \mathbb{R}_*^+$ denotes a temperature parameter modulating the sharpness of the sampling distribution, $e_{g_i}$ denotes the one-hot encoding of $g_i$, and $\odot$ denotes the Hadamard product that is used with $e_{g_i}$ to mask out the $g_i$th prototype and thereby enforce $\tilde{g}_i \neq g_i$.

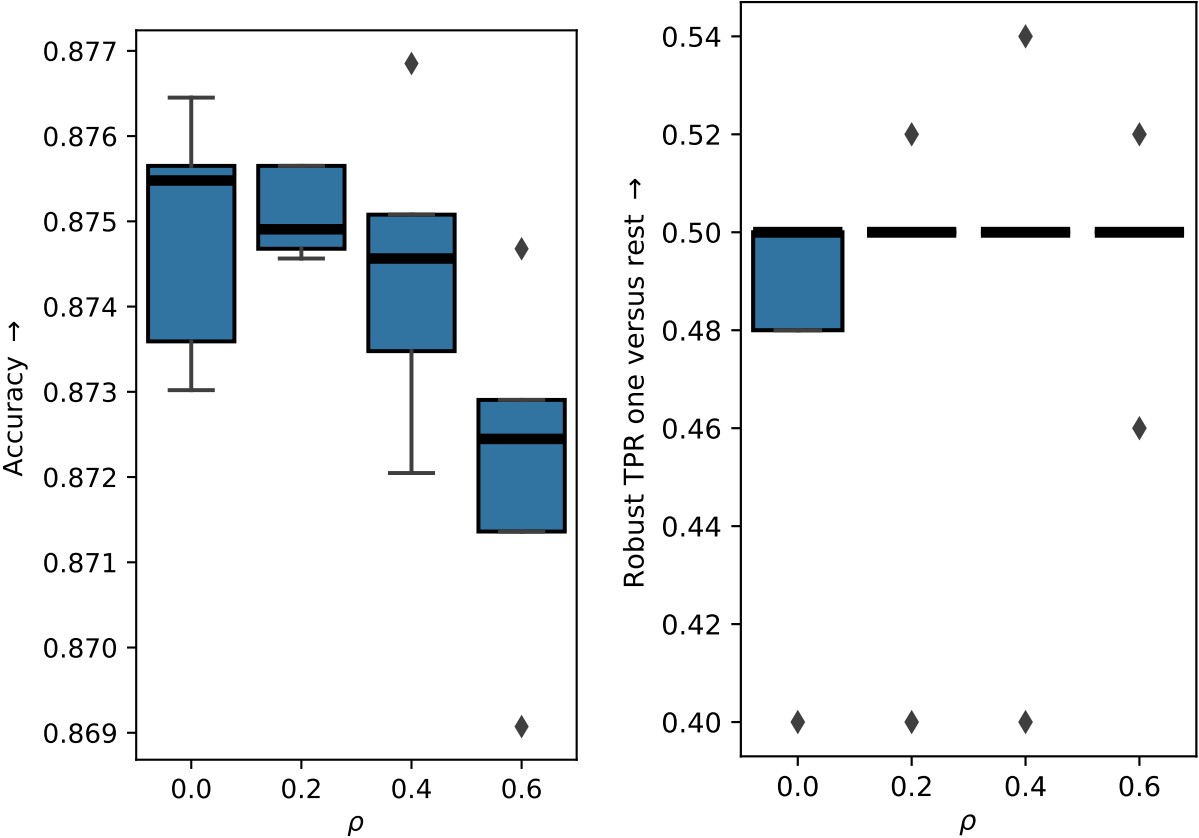

Figure 18: Results from the label-noise sensitivity analysis conducted on the NICO++ dataset. To assay the robustness of our support-matching algorithm to different levels of bag-balance, we run the algorithm with $\rho \in \{0\%, 20\%, \ldots, 60\%\}$ (five replicates for each) of the ground-truth deployment-set labels perturbed (flipping a given label $g_i$ to another value $\tilde{g}_i \in \mathcal{G} \setminus \{g_i\}$) according to the scheme outlined in Appendix J, with all other factors held constant; $\rho = 0$ corresponds to `Ours (Oracle=B)`, i.e., our method with the bag-balancing oracle, as featured in the main set of experiments in Sec. 4.3. We observe that the algorithm is stable across the range of $\rho$ values in terms of both `Robust Accuracy` (**left**) and `Robust TPR one versus rest` (**right**).

