# OpenReview forum: "Addressing Attribute Bias with Adversarial Support-Matching"
_TMLR — Accepted by TMLR_

### Review · Reviewer_c3BS · 2023-12-03

**Summary Of Contributions:**

This paper considers a certain case of selection bias where the selection is dependent on the joint distribution of label class and sensitive attribute. Hereby, the assumption is that the sensitive attribute is can be classified by the covariate variables though independent of the labels. The authors call this problem as attribute bias. To address this issue, the authors propose a solution consisting of encoder, decoder, and discriminator, and developed the adversarial support matching algorithm. The authors demonstrate the effectiveness of the approach on three vision datasets.

**Audience:**

Yes

**Claims And Evidence:**

Yes

**Requested Changes:**

1. The experiment is conducted only on the image datasets. I would suggest the authors at least work on one tabular dataset.
2. Please report the joint distribution of class and sensitive attribute for the Coloured MNIST and CelebA datasets.
3. Try to evaluate the standard fairness metrics in the experimental results, including demographic parity, equality of opportunity, etc.

**Strengths And Weaknesses:**

Strengths:
The authors solved an intriguing yet important problem, where the selection bias is an effect of both label class and sensitive attribute. As far as I know, this setting is less touched in the fairness literature. The paper is clearly written in general, with a full description on formulations and algorithms.

Weaknesses:
1. The proposed solution requires the access to a deployment set, which is typically unavailable in the real world applications. The assumption that the sensitive attribute is independent of label class but can be predicted is too strong.
2. The proof is not straightforward. In Eq. (26), how did you eliminate the first term and directly sum up $\tilde{O}(\sqrt{1 / N})$ in Eq. (27)?
3. It confused me a lot. How necessary is it to apply the attention mechanism in the discriminator module? Is the attention more favorable than the other model architectures?

---

> ### Author Response · Authors · 2024-01-06
>
> We thank you kindly for your review. We respond below to the individual points raised in your review.
>
> **Weaknesses**
>
> 1.  *The proposed solution requires the access to a deployment set, which is typically unavailable in real-world applications. The assumption that the sensitive attribute is independent of label class but can be predicted is too strong.*
>
>     We would argue that the availability of an *unlabelled* dataset derived from the target population is not an overly strong assumption; this same assumption is the basis of Domain Adaptation, for instance, and one can imagine a dataset of this nature being obtainable from sources such as census data -- as alluded to in the main text, one may even embrace a transductive approach under appropriate conditions and use the test set as the deployment set, thereby side-stepping the assumption altogether. We consider a setup in which the sensitive attribute is easier to predict than the class label and spuriously correlated with it. If this were not the case, then ERM would work just fine.
>
> 2.  *The proof is not straightforward. In Eq. (26), how did you eliminate the first term and directly sum up $\tilde{O}(1/N)$ in Eq. (27)?*
>
>     We apologize for the confusion. The first term is not eliminated; as the term we can estimate, the question is only how large the difference is between this term and the term that we theoretically want. We have expanded this step of the proof in the revised manuscript.
>
> 3.  \[...\] *How necessary is it to apply the attention mechanism in the discriminator module? Is the attention more favourable than the other model architectures?*
>
>     Fig. 4 (bottom) shows the results of an ablation study where the aggregator (the attention module) is the ablated factor, from which we can observe the benefit of casting the problem as one of support-matching instead of distribution matching. If the question is not about the use of aggregation but the choice of the function performing it, which is to say the benefit of attention-based aggregation over non-parametric aggregation (most typically uniform-weighting), then we would point to past literature evidencing this (Ilse et al., 2018; Lee et al., 2019) as well as the context-specific intuition (grounded by Fig. 11) that the discriminator needs to learn to selectively attend to those samples associated with the missing groups to distinguish between the training and deployment sets.
>
> **Requested changes**
>
> 1.  *The experiment is conducted only on the image datasets. I would suggest the authors at least work on one tabular dataset.*
>
>     We have not included experiments on tabular datasets so far because invariant representations are less interpretable there (meaning analysis like in Figure 10 in the Appendix) and because they would require significant effort to set up the reconstruction loss correctly. Prior work like Yang et al. (2023) also did not study tabular datasets (only image and text datasets). We already have fairly extensive experiments with 3 quite different image datasets, so we would not see additional experiments on tabular data as vital, and would leave it for future work. We have added text to the revised paper to clarify the modality of the datasets used and to mention other modalities as avenues for future work.
>
> 2.  *Please report the joint distribution of class and sensitive attributes for the Coloured MNIST and CelebA datasets.*
>
>     For Coloured MNIST, this information is derivable from the biasing matrix in Appendix G.1, noting that each class in the full (train and test splits combined) dataset is represented by $7$K examples and 20% and 40% of that dataset are withheld for deployment (which is further subsampled, group-conditionally, according to the matrix) and testing, respectively. We have revised the manuscript to include statistics for the CelebA dataset; these can be found in Tables 9-12 in Appendix G.2. We apologise for the omission.
>
> 3.  *Try to evaluate the standard fairness metrics in the experimental results, including demographic parity, equality of opportunity, etc.*
>
>     We note at the beginning of Sec. 4 the inclusion of additional plots in Appendix H.6 showing the TPR ratios (corresponding to equality of opportunity), TNR ratios, and Positive Rate ratios (corresponding to demographic parity) associated with the results presented in Sec. 4 for the Coloured MNIST and CelebA datasets. For NICO++, we cannot compute conventional binary fairness metrics due to both the attribute and target being multiclass.
>
> **References**
>
> Ilse, M., Tomczak, J., & Welling, M. (2018). Attention-based deep multiple instance learning. ICML.
>
> Lee, J., Lee, Y., Kim, J., Kosiorek, A., Choi, S., & Teh, Y. W. (2019). Set transformer: A framework for attention-based permutation-invariant neural networks. ICML.
>
> Yang, Y., Zhang, H., Katabi, D., & Ghassemi, M. (2023). Change is Hard: A Closer Look at Subpopulation Shift. ICML.

---

### Review · Reviewer_qqcC · 2023-12-12

**Summary Of Contributions:**

This paper challenges a specific spurious correlation problem variant involving missing groups or (attribute label, class label) pairs. Specifically, the spurious correlation problem considers the presence of some secondary attribute correlated to the class label in a spurious way. In particular, if some attribute is completely missing for some label, the machine learning algorithm might learn the correlation between the absence of the attribute and the label. The authors mainly consider the case that such a missing group exists in the training set.

Based on an adversarial setting, the authors propose a new method, Adversarial Support Matching. To overcome the missing groups, the authors introduce a deployment set that contains only unlabeled samples but contains all groups. The authors first reduce the given problem to an optimization problem that requires matching distributions between the training set and the deployment set over all (attribute label, class label) pairs. Proposition 1 provides the rationale for the authors’ objective (Equation 6).

In the actual implementation, the authors convert the objective function to a loss function with two terms: a reconstruction loss of the encoder-decoder pair and the loss from the distribution matching using an adversarial discriminator function. The loss function is computed over bags of samples, and the discriminator’s goal is to distinguish where the bags are from based on the encoder’s output. Therefore, being able to deceive the discriminator means that the encoder induces a similar distribution regardless of whether the input is from the training or deployment set.

Choosing bags representing the training set / the deployment set is based on semi-supervised clustering algorithms. Proposition 2 provides the error bound for the clustering-based bag balancing.

With experiments, the authors demonstrate the performance of the proposed method. The authors compared their method to three baseline methods on three datasets: Coloured MNIST, CelabA, and NICO++. On all three datasets, the proposed method outperformed the baseline methods.

**Audience:**

Yes

**Broader Impact Concerns:**

I don’t see a particular broader impact concern regarding this paper.

**Claims And Evidence:**

Yes

**Requested Changes:**

Consider addressing the following minor issues regarding the paper writing.
  - One small typo in Section 2.1. In the sentence “for $Y=1$, however, we only observe samples form the negative class.”, $Y=1$ already means the positive class. This should be $A=1$.
  - Both Figure 8 and Algorithm 1 are in the Appendix. Clearly state that Figure 8 and Algorithm 1 are in the Appendix, e.g., in the first sentence of Section 3.3.
  - Provide tables summarizing experimental results in the Appendix. While charts are good visualizations of the experimental results, people would be interested in a more specific quantity.
  - If I understood correctly, any result for a nontrivial oracle is practically not plausible but the authors presented the results as upper bounds. This might be misleading people and could provide some wrong impression about the performance. I’d suggest putting all box plots for nontrivial oracles at the bottom of the chart and draw a line separating trivial oracle results from the upper bound results.

**Strengths And Weaknesses:**

# Strengths
  1. The authors approach the problem in a very creative way.
  2. This work contains theoretical arguments with formal proofs for the statements. Those formal analyses of the method add more value to this paper.
  3. The authors used various experimental settings. Specifically, the authors compared their work to three existing baseline methods on three different datasets.
     - In particular, the datasets cover both simple data (Coloured MNIST) and more complicated data (CelebA and NICO++), including the scenario of a large number of groups.
  4. The experimental results show some improvements against the baseline.

# Weaknesses
  1. As mentioned in the problem setup, this paper tackles a specific variant of the spurious correlation (SC) problem involving missing groups. However, I’m not convinced enough about the importance of this variant. Is the scenario with missing groups common in practice?
     - One crucial question is whether or not the proposed method can be generalized, e.g., in a setting where some groups are very scarce rather than missing. Can the authors provide some thoughts on how to generalize the proposed method?
  2. If the training set has missing groups, we also need to doubt the assumption of a group-complete deployment set.
     - Can we argue that we have a high probability group-complete deployment set if we have sufficient random samples?

---

> ### Author Response · Authors · 2024-01-06
>
> We thank you kindly for your review. We respond below to the individual points raised in your review.
>
> **Weaknesses**
>
> 1.  *As mentioned in the problem setup, this paper tackles a specific variant of the spurious correlation (SC) problem involving missing groups. However, I'm not convinced enough about the importance of this variant. Is the scenario with missing groups common in practice? One crucial question is whether or not the proposed method can be generalized, e.g., in a setting where some groups are very scarce rather than missing. Can the authors provide some thoughts on how to generalize the proposed method?*
>
>     The problem of missing groups was previously mentioned in Yang et al. (2023) (without a working solution and under the name of "Attribute Generalization"), which is some indication that the problem is of interest.
>
>     Here are our thoughts on how to generalize the proposed method to the case where some groups are very scarce: Samples from low-sample groups would, perhaps counterintuitively, be placed in the "deployment set" (the unlabelled contrastive dataset). That way, they would be expected to enhance the contrast between the training set and the deployment set, such that the signal for the invariance learning is stronger. The labels of these samples from the low-sample groups can still be used to generate more balanced bags (e.g., by using them as side-information for clustering). The alternative of putting samples from low-sample groups into the "training set" has the disadvantage that the model could overfit on them if we attempt to group-balance the batches.
>
> 2.  *If the training set has missing groups, we also need to doubt the assumption of a group-complete deployment set. Can we argue that we have a high probability group-complete deployment set if we have sufficient random samples?*
>
>     We assume the deployment set has support matching that of the test set and it is w.r.t. this that our notion of group-completeness is defined, not w.r.t. some universal set, i.e. one defining all possible values the group might take (all nationalities, for instance, rather than just those nationalities contained within the test set). As such, our objective is a much more grounded one and the completeness can, for example, be trivially satisfied through a transductive setting in which the deployment and test sets are one and the same.
>
> **Requested Changes**
>
> We have hopefully addressed all of the reviewer's requested changes. These (along with all other revisions) have been highlighted in red for salience. The tables summarising the experimental results can be found in Appendix C.
>
> **References**
>
> Yang, Y., Zhang, H., Katabi, D., & Ghassemi, M. (2023). Change is Hard: A Closer Look at Subpopulation Shift. *Proceedings of the 40th International Conference on Machine Learning.*

---

### Review · Reviewer_zet4 · 2023-12-17

**Summary Of Contributions:**

This paper proposes to address the attribute bias, i.e. the shortcut learning problem, by using an additional diverse deployment dataset. The deployment set is assumed to cover all the pairs of attributes and labels. Balanced bags are sampled from the training and deployment sets using semi-supervised clustering. Then an encoder is trained in an adversarial manner to produce features that are indistinguishable in terms of which set it is from.

**Audience:**

Yes

**Claims And Evidence:**

No

**Requested Changes:**

Please refer to the weaknesses above.

- Add a detailed description of the training set, the deployment set, and the test set.
- Add related ablation studies.
- Add analysis on the impact of the semi-supervised clustering.

**Strengths And Weaknesses:**

Strengths:
1. The paper is well-written and the problem is well formulated.
2. The proposed method is generally interesting and the adversarial training is intuitively reasonable.

Weaknesses:
1. In the experiments, the detailed training set, the deployment set, and the test set are not clearly described.
2. The deployment set is assumed to be group-complete. This assumption might be too strong for practical use.
3. The ablation on the model components (f,h,t,r) and the loss terms (reconstruction, match) are not provided.
4. The semi-supervised clustering is critical in the proposed method. The impact of the performance of clustering algorithms should be studied.
5. In Eq.9, do we need to use paired data for x \in bdep and x \in btr? Do these two x have the same class label?

Minors:
"is missing groups" > "are missing groups"

---

> ### Author Response · Authors · 2024-01-06
>
> We thank you kindly for your review. We respond below to your points in the "weaknesses" section.
>
> 1.  *In the experiments, the detailed training set, the deployment set, and the test set are not clearly described.*
>
>     For Coloured MNIST, this information is derivable from the biasing matrix in Appendix G.1, noting that each class in the full (train and test splits combined) dataset is represented by $7$K examples and 20% and 40% of that dataset are withheld for deployment (which is further subsampled, group-conditionally, according to the matrix) and testing, respectively. We have revised the manuscript to include statistics for the CelebA dataset; these can be found in Tables 9-12 in Appendix G.2. We apologise for the omission. The splits for the NICO++ dataset are constructed according to Yang et al. (2023), with the validation set playing the part of the deployment set.
>
> 2.  *The deployment set is assumed to be group-complete. This assumption might be too strong for practical use.*
>
>     We assume the deployment set has support matching that of the test set and it is w.r.t. this that our notion of group-completeness is defined, not w.r.t. some universal set, i.e. one defining all possible values the group might take (all nationalities, for instance, rather than just those nationalities contained within the test set). As such, our objective is a much more grounded one and the completeness can, for example, be trivially satisfied through a transductive setting in which the deployment and test sets are one and the same.
>
> 3.  *The ablation on the model components (f,h,t,r) and the loss terms (reconstruction, match) are not provided.*
>
>     While we cannot ablate $f$ as this is the model we are learning for downstream classification, we have conducted ablations on the $r$ and $c$ components, corresponding to the reconstruction loss and supervised (cross-entropy) classification (w.r.t. $y$) loss, respectively. The results of these can be found in Fig. 16 in Appendix I.3 in the newest revision of the paper. We note that we had already conducted ablations on the $h$ (the discriminator) w.r.t. its aggregator (instance-wise vs bag-wise), the results of which can be found in Fig. 4 (bottom).
>
> 4.  *The semi-supervised clustering is critical in the proposed method. The impact of the performance of clustering algorithms should be studied.*
>
>     We note that we have already conducted a fairly comprehensive study of the influence of the quality of the clustering on the support matching algorithm in the form of a sensitivity analysis, with the sensitivity in question being measured w.r.t. the correctness of the cluster labels. The results of this study can be found in Appendix K. Our sensitivity analysis simulates different levels of correctness of the clustering, so there is no need to try out different clustering algorithms. Our method would benefit from improved clustering methods, but the evaluation and improvement of clustering methods do not fall within the paper's scope.
>
> 5.  *In Eq.9, do we need to use paired data for x \\in bdep and x \\in btr? Do these two x have the same class label?*
>
>     Individual samples do not have to be paired according to class label. (As the deployment set is unlabelled, this wouldn't actually be possible.) Rather, we try to ensure that (in expectation) all classes are present in the same proportion in all bags. So, the data is only paired on the "bag-level" and not the sample level. This pairing then empirically suffices to learn the correct invariance.
>
> **References**
>
> Yang, Y., Zhang, H., Katabi, D., & Ghassemi, M. (2023). Change is Hard: A Closer Look at Subpopulation Shift. *Proceedings of the 40th International Conference on Machine Learning.*

---

### Review · Reviewer_mSE2 · 2023-12-17

**Summary Of Contributions:**

This paper targets a challenging data-absent scenario where the absence of certain data is linked to the second level of a two-level hierarchy in the data. To address this, this paper proposes an adversarial support-matching method invariant to the spurious correlations. Experimental results demonstrate the effectiveness of the proposed method. Theoretical analysis proves that the proposed method can yield representations invariant to attributes.

**Audience:**

Yes

**Claims And Evidence:**

Yes

**Requested Changes:**

The authors should address all my concerns in weaknesses.

**Strengths And Weaknesses:**

Strength:

The motivation and the contribution are clearly stated.

Theoretical and experimental studies support the effectiveness of the proposed method.


Weakness:

1.	The primary concern is the encoder (f) assumption in proposition 1. It requires that the conditional distributions on the training and deployment sets are identical for all attributes and classes, which might be challenging to meet in practice (such as the OOD scenario). The authors should present more discussions and empirical studies about this. Besides, it is advisable to generalize the results to a more general case such as measuring their distribution gap by a small positive value $\epsilon$.

2.	Some statements are confusing and need to be clarified. For instance, in sec.3.1, “if the class y' has incomplete a'-support, it should be referred to as a' support rather than a-support.” Here, the concept of “a distribution over an empty set for some attributes” requires further clarification. In Equation (4), it should be “y’ has full a-support” rather than “y has full a-support”. In Equation (5), If there is more than one element in this set Π(a′ ,y′ ), which one should be chosen?

3.	This work is highly related to the hierarchical classification problems [1,2]. It is essential to comprehensively analyze and compare related works, methodologies, and experiments to provide a complete study.

4.	Another concern is that the reviews of related work are insufficient. In unsupervised domain adaptation and multiple instance learning paragraphs, the authors overlook some state-of-the-art studies from the AUC optimization perspective, such as [5, 6].

5.	In experiments, the authors merely study the effectiveness of the proposed method over balanced datasets. Considering that the real-world data usually follows a long-tail property, it is advisable to conduct experiments on some imbalanced cases. Besides, I noticed that [3,4] started an early trial to explore the effective DRO approaches under the long-tail distribution, which achieves promising results. It is essential to consider these methods in the experiments.

Reference

[1] Jack Valmadre. Hierarchical classification at multiple operating points. Advances in Neural Information Processing Systems. 2022.

[2] J. Chen, P. Wang, J. Liu and Y. Qian, "Label Relation Graphs Enhanced Hierarchical Residual Network for Hierarchical Multi-Granularity Classification," 2022 IEEE/CVF Conference on Computer Vision and Pattern Recognition, doi: 10.1109/CVPR52688.2022.00481.

[3] D. Zhu, G. Li, B. Wang, X. Wu, and T. Yang. When auc meets dro: Optimizing partial auc for deep learning with non-convex convergence guarantee. In International Conference on Machine Learning, pages 27548–27573. PMLR, 2022.

[4] Dai, Siran and Xu, Qianqian and Yang, Zhiyong and Cao, Xiaochun and Huang, Qingming. DRAUC: An Instance-wise Distributionally Robust AUC Optimization Framework. Advances in Neural Information Processing Systems. 2023.

[5] Z. Yang et al., "AUC-Oriented Domain Adaptation: From Theory to Algorithm," in IEEE Transactions on Pattern Analysis and Machine Intelligence, vol. 45, no. 12, pp. 14161-14174, Dec. 2023, doi: 10.1109/TPAMI.2023.3303943.

[6] D. Zhu et al., “Provable Multi-instance Deep AUC Maximization with Stochastic Pooling,” in International Conference on Machine Learning, 2023.

---

> ### Author Response · Authors · 2024-01-06
>
> We thank you kindly for your review. We respond below to your points in the "weakness" section.
>
> 1.  *The primary concern is the encoder (f) assumption in proposition 1. It requires that the conditional distributions on the training and deployment sets are identical for all attributes and classes, which might be challenging to meet in practice (such as the OOD scenario). The authors should present more discussions and empirical studies about this. Besides, it is advisable to generalize the results to a more general case such as measuring their distribution gap by a small positive value.*
>
>     We do not assume any shift in the conditional distribution, i.e. we have that $P\left( X^{\operatorname{tr}}|G^{\operatorname{tr}} \right) \approx P\left( X^{\text{te}}|G^{\operatorname{tr}} \right)$, with the attribute-bias problem stemming only from a mismatch in the marginal distributions $P\left( G^{\text{tr}} \right)$ and $P\left( G^{\text{te}} \right)$. We would leave it to future work to explore the intersectional problem of attribute bias under distribution shift; to introduce a novel problem and non-trivial extensions of it would go beyond the scope of a single paper.
>
> 2.  *Some statements are confusing and need to be clarified.* \[...\]
>
>     The listed points of confusion have been amended in the latest revision of the paper; thank you for bringing them to our attention. If there are other instances, please let us know.
>
> 3.  *This work is highly related to the hierarchical classification problems \[1,2\]. It is essential to comprehensively analyze and compare related works, methodologies, and experiments to provide a complete study.*
>
>     While we are committed to drawing connections with adjacent fields, we would argue that the problem of hierarchical classification is quite distinct from that of the attribute bias considered herein. Our goal is not to predict, in addition to the superclass (class) the subclass (attribute), on the contrary, our goal is to achieve *invariance* to it since it serves as a confounding factor. Although the problem has a (two-level) hierarchical aspect to it, in how the data is biased, the prediction task defined on it is not itself hierarchical: it is only the top level of the two-level hierarchy which we want to predict. We have added a brief discussion of hierarchical classification to Appendix B (Related Work).
>
> 4.  *Another concern is that the reviews of related work are insufficient. In unsupervised domain adaptation and multiple instance learning paragraphs, the authors overlook some state-of-the-art studies from the AUC optimization perspective, such as \[5, 6\]*
>
>     While our paper certainly takes inspiration from multiple-instance learning (MIL) we would argue that the paper is not one *of* or *about* MIL – the goal is not to optimise for multiple-instance classification but for attribute-invariance, using MIL as an engine for support alignment, finding it to outperform instance-wise discrimination. As such, we are not convinced a review of recent MIL literature is of great merit; it may be that we can improve our method by incorporating ideas from the given references but they do not alter the core of it. Thank you for bringing the work on AUC-optimization to our attention. However, it does not seem directly relevant to the work. If the reviewers feel strongly otherwise, however, we will gladly expand the related work along the suggested lines.
>
> 5.  *In experiments, the authors merely study the effectiveness of the proposed method over balanced datasets. Considering that the real-world data usually follows a long-tail property, it is advisable to conduct experiments on some imbalanced cases. Besides, I noticed that \[3,4\] started an early trial to explore the effective DRO approaches under the long-tail distribution, which achieves promising results. It is essential to consider these methods in the experiments.*
>
>     Understanding the reviewer to mean that the experimented-with datasets are *balanced* in the sense of having an approximately-uniform distribution of groups within the training set, this is very much not the case, as can be seen from the dataset statistics we have added in Appendix G.2 of the revised manuscript (per the request of reviewers `c3BS`/`zet4`). The problem we tackle is distinct from long-tail learning since it is not a matter of certain classes being underrepresented but rather altogether absent, such that the methods contained in the given references, and pre-existing methods generally, are not suitably equipped to handle it.
>
>     As we understood the papers \[3,4\], both utilize DRO for AUC-optimization, but do not propose improvements to or generalisations of the DRO algorithm.

---

### Author Response · Authors · 2024-01-06
**New revision**

Happy New Year!

We thank all reviewers for their comments. We have uploaded a revised version of the paper that we hope represents a meaningful improvement and allays some of the raised concerns, in conjunction with our responses here. We have specified in the reviewer-specific responses where any relevant revisions have been made. All changes in the paper are highlighted in (dark-ish) red.

---

### Decision · Action_Editor_WiJw · 2024-02-01

**Recommendation:** Accept as is

**Comment:**

This paper proposes Adversarial Support Matching, an approach to address a specific spurious correlation problem where certain attributes are completely missing for some labels in the training set. The proposed method introduces a deployment set to learn a representation that is invariant to the attributes. Reviewers generally acknowledge the significance of the research problem while raising several concerns, such as strong assumptions of the missing group in the training dataset (qqcC, c3BS, zet4) and somehow blurred description in method and experiment setting (mSE2, zet4). During rebuttal, the authors successfully resolved most of the concerns, and most of the reviewers (qqcC, zet4, mSE2) believe the paper meets the requirements to be published and recommend acceptance. Reviewer c3BS believes the authors could have done a better job at comparing the tabular data and the evaluation for other group fairness metrics. I hope authors can make improvements in their camera-ready version.

**Audience:**

Yes.

**Claims And Evidence:**

Yes.